# Adaptive Kalman Filter with $L_2$ Feedback Control for Active Suspension Using a Novel 9-DOF Semi-Vehicle Model

**Huan Yang** **, Jiang Liu \*, Min Li, Xilong Zhang, Jianze Liu and Yulan Zhao**

School of Mechanical and Automotive Engineering, Qingdao University of Technology, Qingdao 266520, China; yanghuan@qut.edu.cn (H.Y.); minli@qtech.edu.cn (M.L.); zhangxilong@qut.edu.cn (X.Z.); liujianze@qut.edu.cn (J.L.); yulanzhao@qtech.edu.cn (Y.Z.)
\* Correspondence: liujiang@qut.edu.cn

**Abstract:** In order to further improve driving comfort, this paper takes the semi-vehicle active suspension as the research object. Furthermore, combined with a 5-DOF driver-seat model, a new 9-DOF driver seat-active suspension model is proposed. The adaptive Kalman filter combined with $L_2$ feedback control algorithm is used to improve the controller. First, a discrete 9-DOF driver seat-active suspension model is established. Then, the $L_2$ feedback algorithm is used to solve the optimal feedback matrix of the model, and the adaptive Kalman filter algorithm is used to replace the linear Kalman filter. Finally, the improved active suspension model and algorithm are verified through simulation and test. The results show that the new algorithm and model not only significantly improve the driver comfort, but also comprehensively optimize the other performance of the vehicle. Compared with the traditional LQG control algorithm, the RMS value of the acceleration experienced by the driver's limb are, respectively, decreased by 10.9%, 15.9%, 6.4%, and 7.5%. The RMS value of pitch angle acceleration experienced by the driver decreased by 6.4%, and the RMS value of the dynamic tire deflection of front and rear tire decreased by 32.6% and 12.1%, respectively.

**Keywords:** active suspension; driver model; $L_2$ state gain feedback control; adaptive Kalman filter



## 1. Introduction

It has been proved that active suspensions can significantly improve the ride comfort of the vehicle [1,2]. The mathematical model and control algorithm are two hot spots in active suspension researches.

At present, the research [3–6] on active suspension is mainly based on the simplified mathematical model, which the driver and seat are unified as the sprung mass for calculation. Bououden [7] took the traditional 1/4 suspension model as the research object and realized the multivariable predictive control of the active suspension system by using the Takagi–Sugeno fuzzy method. However, under real working conditions, the actuator would act on a complex multi-DOF system. The simplification of the traditional models makes the description on this "people–vehicle–road" system unsatisfied. Bouazara [8] improved the traditional 1/4 model into a 3-DOF model, took into account the feedback of seats, and adopted the sequential unconstrained minimization technology to reduce the vibration acceleration transmitted from the road surface to the driver. The works in [9,10] also conducted research on the basis of the 3-DOF model. Afterwards, Hu [11] improved the 3-DOF model and used fuzzy PID integrated control strategy to control the 8-DOF seat suspension model of the whole vehicle. The control results show that the improved model and control strategy can greatly improve the ride comfort and handling stability. However, the above models still do not really consider the driver's sensory experience. Regarding the human body as a rigid body cannot accurately reflect the biodynamic characteristics of the human body. Sever [12] further improved the mathematical model of suspension. Based on the 1/4 seat-active suspension model, a 2-DOF driver model that contains two degrees of freedom in the upper and lower limbs was introduced, and the riding comfort was greatly

improved with the multi-state variable feedback of the driver, seat, and active suspension. After that, the works in [13,14] continuously refined the driver models to make the model more realistic. However, these models are all researched on the basis of the 1/4 suspension model. The driver's experience is limited to vertical vibration, and it is difficult to fully evaluate the control response of the driver. It is known that people are more sensitive to axial and lateral vibration, and the influence of the pitch of the body and the vibration difference between front and rear axle on the driver cannot be ignored either. Therefore, it is necessary to establish a new mathematical model that provides driver with sufficient vibration input information and output performances.

In terms of control algorithm, Zhao [15] used particle swarm optimization algorithm to optimize the parameters of PID controller. The deviation between the actual acceleration value and the initial acceleration value was taken as the control input, and the ride comfort was improved through PID closed-loop feedback. The works in [16,17] also used a PID control algorithm to improve the ride comfort. However, the above study only conducted feedback control on the body acceleration, and other performance indicators of the model, such as SWS and DTD, were not optimized, and the control effect was not ideal. Therefore, Lan [18] applied the LQG algorithm to the traditional vibration model and solved the optimal feedback matrix through the Riccati equation. Finally, the stability and ride comfort were optimized by multi-parameter feedback control. The works in [19–21] also used LQG control algorithm and optimized the weighted coefficients of performance indexes through genetic algorithm and other optimization algorithms. However, the LQG control algorithm needs to measure too many states and the states are difficult to measure, which limits the application of the LQG control algorithm. For this reason, Yu [22] combined LQG control with linear Kalman filter algorithm to achieve filtering estimation of all controlled states by measuring BA, SWS, and DTD. Zhu [23] used Kalman filter algorithm and realized the fault detection of active suspension under finite frequency domain by the generalized Kalman–Yakubovich–Popov algorithm. On this basis, the Kalman filter was also used to improve the control algorithm in [24–27]. However, in order to satisfy the positive definite quadratic equation, the suspension model must be simplified. After introducing the driver model, it is difficult for the system state matrix to meet this condition due to the increase of the model's degree of freedom. In addition, the time-varying noise variance will reduce the accuracy of the linear Kalman filter [28,29]. These two main reasons make the traditional linear Kalman filter almost impossible to be directly applied to the driver-active suspension model. Therefore, we must find an adaptive method that can reduce noise and enlarge the DOF of the model.

Therefore, a novel 9-DOF discrete model is proposed. The model consists of a 4-DOF driver model, a seat, and a 4-DOF semi-vehicle active suspension model. The new model increases the influence of pitch motion on the driver. Then, an $L_2$ gain feedback control is used to give the control matrix $K$ without positive definite quadratic equation, and an adaptive Kalman algorithm is designed to replace the linear Kalman filter. Put the matrix $K$ and the responses of the vehicle and the driver into the new filter, and the control force of the actuator is the output. The new model and algorithm can realize the real-time multi-parameter optimal feedback control. Finally, the improved model and algorithm are verified by simulation and test.

## 2. Driver-Active Suspension Model

### 2.1. Suspension Control Principle

On the basis of the semi-car suspension model, the driver model and the $L_2$ feedback Kalman filter algorithm are applied to the active suspension. The full-text control principle is shown in Figure 1. The $L_2$ feedback algorithm solves the linear matrix inequality to obtain the optimal feedback gain vector according to the established multi-DOF model. Moreover, it is substituted into the adaptive Kalman filter to achieve multi-parameter control of driver performance indexes and vehicle performance indexes.

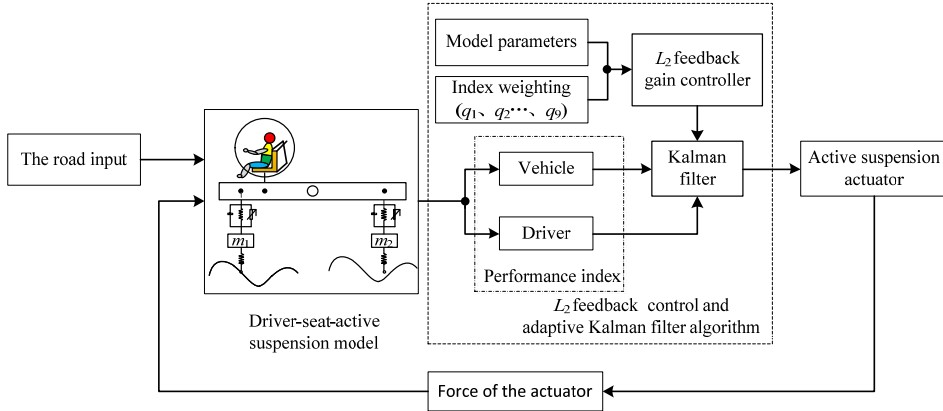

**Figure 1.** Schematic diagram of $L_2$ feedback Kalman filter algorithm for 9-DOF semi-vehicle model.

In the above figure, the analysis of driver seat-active suspension is in Section 2 of this paper, the analysis of $L_2$ feedback control is in Section 3 of this paper, and the adaptive Kalman filtering algorithm is in Section 4 of this paper.

### 2.2. Semi-Vehicle Driver-Active Suspension Model

In vehicle suspension research, the human body is usually regarded as a sprung mass. However, the simplified model cannot really consider the driver's sensory experience. It will not accurately reflect the biodynamic characteristics of the human body. Therefore, when modeling active suspension, it is necessary to introduce a complex biomechanical model of the driver to improve the driver comfort. The 9-DOF semi-vehicle driver-active suspension model established in this paper is shown in Figure 2, where $m$ is sprung mass, that is, the mass of half the car body, and $m_1$ and $m_2$ are the unsprung mass of the front tire and the rear tire, respectively. The seat-driver model divides the seat and the driver's torso into five parts, namely seat mass $m_3$, hip and thigh mass $m_4$, waist mass $m_5$, chest mass $m_6$, and head and neck mass $m_7$.

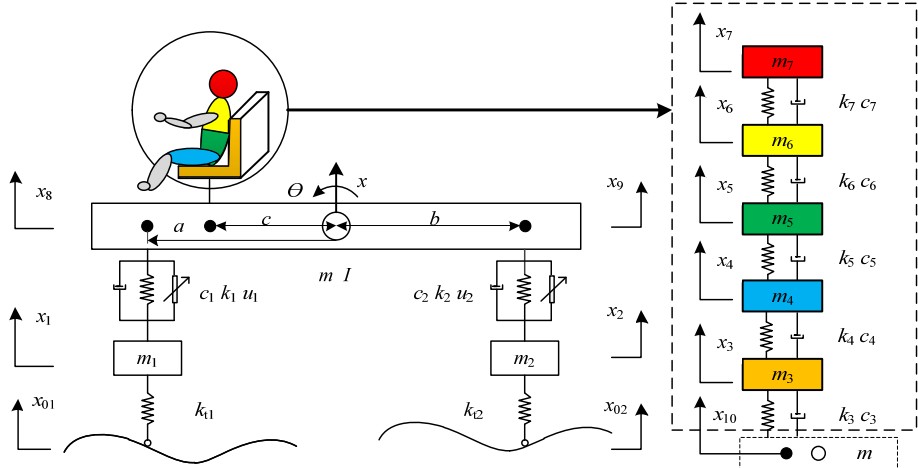

**Figure 2.** Semi-vehicle driver-active suspension model.

As shown in Figure 2, the quality of each part is analyzed by Newton's second law and establish their respective mechanical equations. The motion equation of unsprung mass of the front axle $m_1$ and unsprung mass of the rear axle $m_2$ can be expressed as

$$m_1\ddot{x}_1 + k_1(x_1 - x_8) + k_{t1}(x_1 - x_{01}) + c_1(\dot{x}_1 - \dot{x}_8) + u_1 = 0, \tag{1}$$

$$m_2\ddot{x}_2 + k_2(x_2 - x_9) + k_{t2}(x_2 - x_{02}) + c_2(\dot{x}_2 - \dot{x}_9) + u_2 = 0, \tag{2}$$

where $u_1$ is the active control force of the front axle actuator, $u_2$ is the active control force of the rear axle actuator, $c_1$ is the suspension damping coefficient of the front axle, $c_2$ is the suspension damping coefficient of the rear axle, $k_1$ is the suspension stiffness of the front axle, $k_2$ is the suspension stiffness of the rear axle, $k_{t1}$ is the tire stiffness of the front tires, $k_{t2}$ is the tire stiffness of the rear tires, $x_1$ is the displacement of the front tires, $x_2$ is the displacement of the rear tires, $x_8$ is the body displacement of the front axle, $x_9$ is the body displacement of the rear axle, $x_{01}$ is the road input of the front tire, and $x_{02}$ is the road input of the rear tire.

The mass of each part of the seat-driver model is analyzed, respectively, and the motion equation of the seat–driver system can be expressed as

$$m_3\ddot{x}_3 + k_4(x_3 - x_4) + c_4(\dot{x}_3 - \dot{x}_4) + k_3(x_3 - x_{10}) + c_3(\dot{x}_3 - \dot{x}_{10}) = 0, \tag{3}$$

$$m_4\ddot{x}_4 + k_5(x_4 - x_5) + c_5(\dot{x}_4 - \dot{x}_5) + k_4(x_4 - x_3) + c_4(\dot{x}_4 - \dot{x}_3) = 0, \tag{4}$$

$$m_5\ddot{x}_5 + k_6(x_5 - x_6) + c_6(\dot{x}_5 - \dot{x}_6) + k_5(x_5 - x_4) + c_5(\dot{x}_5 - \dot{x}_4) = 0, \tag{5}$$

$$m_6\ddot{x}_6 + k_7(x_6 - x_7) + c_6(\dot{x}_6 - \dot{x}_7) + k_6(x_6 - x_5) + c_6(\dot{x}_6 - \dot{x}_5) = 0, \tag{6}$$

$$m_7\ddot{x}_7 + k_7(x_7 - x_6) + c_7(\dot{x}_7 - \dot{x}_6) = 0, \tag{7}$$

where $c_3$ is the seat damping coefficient, $c_4$ is the hip and thigh damping coefficient, $c_5$ is the waist damping coefficient, $c_6$ is the chest damping coefficient, $c_7$ is the head and neck damping coefficient, $k_3$ is the seat stiffness, $k_4$ is the hip and thigh stiffness, $k_5$ is the waist stiffness, $k_6$ is the chest stiffness, $k_7$ is the head and neck stiffness, $x_3$ is the displacement of the seat, $x_4$ is the displacement of the driver's hips and thighs, $x_5$ is the displacement of the driver's waist, $x_6$ is the displacement of the driver's chest, $x_7$ is the displacement of the driver's head and neck, and $x_{10}$ is the displacement of the connection between body and seat.

The force balance equation and the torque balance equation around the center of mass are, respectively, established by analyzing the center of mass,

$$m\ddot{x} + k_1(x_8 - x_1) + c_1(\dot{x}_8 - \dot{x}_1) + k_2(x_9 - x_2) + c_2(\dot{x}_9 - \dot{x}_2) + k_3(x_{10} - x_3) + c_3(\dot{x}_{10} - \dot{x}_3) - u_1 - u_2 = 0, \tag{8}$$

$$I\ddot{\theta} - ak_1(x_8 - x_1) - ac_1(\dot{x}_8 - \dot{x}_1) + bk_2(x_9 - x_2) + bc_2(\dot{x}_9 - \dot{x}_2) - ck_3(x_{10} - x_3) - cc_3(\dot{x}_{10} - \dot{x}_3) + au_1 - bu_2 = 0, \tag{9}$$

where $I$ is the moment of inertia of semi-vehicle, $\theta$ is the pitch angle of the vehicle, $a$ is the distance from the center of mass of the vehicle to the front axle, $b$ is the distance from the center of mass of the vehicle to the rear axle, $c$ is the distance from the center of mass of the vehicle to the center of mass of seat, and driver and $x$ is the displacement of center of mass.

According to the relationship of the body structure, when the pitch angle is small, there is approximately,

$$x_8 = x - a\tan\theta \approx x - a\theta, \tag{10}$$

$$x_9 = x + b\tan\theta \approx x + b\theta, \tag{11}$$

$$x_{10} = x - c\tan\theta \approx x - c\theta. \tag{12}$$

Calculating the second derivative on both sides of the Equations (10)–(12), respectively, we can get

$$\ddot{x}_8 = \ddot{x} - a\ddot{\theta}, \tag{13}$$

$$\ddot{x}_9 = \ddot{x} + b\ddot{\theta}, \tag{14}$$

$$\ddot{x}_{10} = \ddot{x} - c\ddot{\theta}. \tag{15}$$

According to Equations (1)–(15), through equation calculation, the semi-vehicle active suspension system can be expressed as

$$\ddot{x}_1 = -\frac{k_1}{m_1}(x_1 - x_8) - \frac{k_{t1}}{m_1}(x_1 - x_{01}) - \frac{c_1}{m_1}(\dot{x}_1 - \dot{x}_8) - \frac{1}{m_1}u_1, \tag{16}$$

$$\ddot{x}_2 = -\frac{k_2}{m_2}(x_2 - x_9) - \frac{k_{t2}}{m_2}(x_2 - x_{02}) - \frac{c_2}{m_2}(\dot{x}_2 - \dot{x}_9) - \frac{1}{m_2}u_2, \tag{17}$$

$$\ddot{x}_8 = -\alpha_1[k_1(x_8 - x_1) + c_1(\dot{x}_8 - \dot{x}_1)] - \alpha_4[k_2(x_9 - x_2) + c_2(\dot{x}_9 - \dot{x}_2)] - \alpha_5[k_3(x_{10} - x_3) + c_3(\dot{x}_{10} - \dot{x}_3)] + \alpha_1 u_1 + \alpha_4 u_2, \tag{18}$$

$$\ddot{x}_9 = -\alpha_4[k_1(x_8 - x_1) + c_1(\dot{x}_8 - \dot{x}_1)] - \alpha_2[k_2(x_9 - x_2) + c_2(\dot{x}_9 - \dot{x}_2)] - \alpha_6[k_3(x_{10} - x_3) + c_3(\dot{x}_{10} - \dot{x}_3)] + \alpha_4 u_1 + \alpha_2 u_2, \tag{19}$$

$$\ddot{x}_{10} = -\alpha_5[k_1(x_8 - x_1) + c_1(\dot{x}_8 - \dot{x}_1)] - \alpha_6[k_2(x_9 - x_2) + c_2(\dot{x}_9 - \dot{x}_2)] - \alpha_3[k_3(x_{10} - x_3) + c_3(\dot{x}_{10} - \dot{x}_3)] + \alpha_5 u_1 + \alpha_6 u_2, \tag{20}$$

where $\alpha_1 = \frac{1}{m} + \frac{a^2}{I}$, $\alpha_2 = \frac{1}{m} + \frac{b^2}{I}$, $\alpha_3 = \frac{1}{m} + \frac{c^2}{I}$, $\alpha_4 = \frac{1}{m} - \frac{ab}{I}$, $\alpha_5 = \frac{1}{m} + \frac{ac}{I}$, $\alpha_6 = \frac{1}{m} - \frac{bc}{I}$.

The seat-driver system can be expressed as

$$\ddot{x}_3 = -\frac{k_4}{m_3}(x_3 - x_4) - \frac{c_4}{m_3}(\dot{x}_3 - \dot{x}_4) - \frac{k_3}{m_3}(x_3 - x_{10}) - \frac{c_3}{m_3}(\dot{x}_3 - \dot{x}_{10}), \tag{21}$$

$$\ddot{x}_4 = -\frac{k_5}{m_4}(x_4 - x_5) - \frac{c_5}{m_4}(\dot{x}_4 - \dot{x}_5) - \frac{k_4}{m_4}(x_4 - x_3) - \frac{c_4}{m_4}(\dot{x}_4 - \dot{x}_3), \tag{22}$$

$$\ddot{x}_5 = -\frac{k_6}{m_5}(x_5 - x_6) - \frac{c_6}{m_5}(\dot{x}_5 - \dot{x}_6) - \frac{k_5}{m_5}(x_5 - x_4) - \frac{c_5}{m_5}(\dot{x}_5 - \dot{x}_4), \tag{23}$$

$$\ddot{x}_6 = -\frac{k_7}{m_6}(x_6 - x_7) - \frac{c_7}{m_6}(\dot{x}_6 - \dot{x}_7) - \frac{k_6}{m_6}(x_6 - x_5) - \frac{c_6}{m_6}(\dot{x}_6 - \dot{x}_5), \tag{24}$$

$$\ddot{x}_7 = -\frac{k_7}{m_7}(x_7 - x_6) - \frac{c_7}{m_7}(\dot{x}_7 - \dot{x}_6). \tag{25}$$

*2.3. Road Input*

The road surface power spectral density is expressed by an approximate fitting formula as

$$G_q(n) = G_q(n_0)\left(\frac{n}{n_0}\right)^{-W}, \tag{26}$$

where $n$ is the spatial frequency, representing the number of wavelengths per unit length; $n_0$ is the reference spatial frequency; $W$ is the frequency index which determines the frequency structure of the power spectral density of road surface; $G_q(n)$ is the spatial frequency power spectral density; and $G_q(n_0)$ is the the road roughness coefficient.

The relationship between spatial frequency and temporal frequency of road roughness is

$$f = vn, \tag{27}$$

$$G_q(f) = \frac{1}{v}G_q(n), \tag{28}$$

where $v$ is the vehicle speed, $f$ is the temporal frequency, $n$ is the spatial frequency, and $G_q(f)$ is the temporal frequency power spectral density.

When the frequency index $W = 2$, the power spectral density of the vertical velocity is

$$G_{\dot{q}}(f) = 4\pi^2 G_q(n_0)n_0^2 v. \tag{29}$$

Road input $x_0$ can be generated by integrating the white noise of Equation (29),

$$\dot{x}_0(t) = 2\pi n_0\sqrt{G_q(n_0)v}w(t). \tag{30}$$

As the spectral density is approximately constant in the low frequency range, the lower cut-off frequency is introduced into the above equation to reflect the road input more truly. Take the Laplace transform of Equation (30) and introduce a lower cutoff frequency $f_0$,

$$G(j\omega) = \frac{2\pi n_0 \sqrt{G_0 v}}{j\omega + \omega_0}. \ (\omega_0 = 2\pi f_0) \tag{31}$$

The inverse Laplace transform of Equation (31) is applied to obtain the time-domain expression of road roughness with the lower cut-off frequency.

$$\dot{x}_0(t) = -2\pi f x_0(t) + 2\pi n_0 \sqrt{G_0 v} w(t), \tag{32}$$

where $w(t)$ is the time domain signal of Gaussian white noise, $\dot{x}_0(t)$ is the time domain signal of road spectrum and $G_0$ is the road roughness coefficient in the international standardization document ISO/TC 108/SC2N67.

Then, for the semi- vehicle model, the road input is

$$\dot{x}_0(t) = -2\pi f_0 x_{01}(t) + 2\pi n_0 \sqrt{G_0 v} w_1(t), \tag{33}$$

$$\dot{x}_0(t) = -2\pi f_0 x_{02}(t) + 2\pi n_0 \sqrt{G_0 v} w_2(t). \tag{34}$$

*2.4. Suspension Modeling*

In order to facilitate subsequent calculations, the differential equations of Equations (19)–(28), (33), and (34) are converted into spatial state expressions in modern control theory. Let the state variable be

$$X = \begin{bmatrix} \dot{x}_1 & \dot{x}_2 & \dot{x}_3 & \dot{x}_4 & \dot{x}_5 & \dot{x}_6 & \dot{x}_7 & \dot{x}_8 & \dot{x}_9 & \dot{x}_{10} & x_1 & x_2 & x_3 & x_4 & x_5 & x_6 & x_7 & x_8 & x_9 & x_{10} & x_{01} & x_{02} \end{bmatrix}^T. \tag{35}$$

Then the spatial state expressions of the semi-vehicle driver's active suspension model is expressed as

$$\dot{X}(t) = AX(t) + BU(t) + GW(t), \tag{36}$$

where

$$A = \begin{bmatrix} A_{11} & A_{12} & A_{13} \\ A_{21} & A_{22} & A_{23} \\ A_{31} & A_{32} & A_{33} \end{bmatrix}$$ is the system state coefficient matrix,

$$B = \begin{bmatrix} -1/m_1 & 0 & 0 & 0 & 0 & 0 & 0 & \alpha_1 & \alpha_4 & \alpha_5 & 0 & 0 & 0 & 0 & 0 & 0 & 0 & 0 & 0 & 0 & 0 & 0 \\ 0 & -1/m_2 & 0 & 0 & 0 & 0 & 0 & \alpha_4 & \alpha_2 & \alpha_6 & 0 & 0 & 0 & 0 & 0 & 0 & 0 & 0 & 0 & 0 & 0 & 0 \end{bmatrix}^T$$ is the system control coefficient matrix,

$$G = \begin{bmatrix} 0 & 0 & 0 & 0 & 0 & 0 & 0 & 0 & 0 & 0 & 0 & 0 & 0 & 0 & 0 & 0 & 0 & 0 & 0 & 0 & 2\pi n_0 \sqrt{G_0 v} & 0 \\ 0 & 0 & 0 & 0 & 0 & 0 & 0 & 0 & 0 & 0 & 0 & 0 & 0 & 0 & 0 & 0 & 0 & 0 & 0 & 0 & 0 & 2\pi n_0 \sqrt{G_0 v} \end{bmatrix}^T$$ is the system noise coefficient matrix,

$U(t) = \begin{bmatrix} u_1(t) \\ u_2(t) \end{bmatrix}$ is the active control vector, $W(t) = \begin{bmatrix} w_1(t) \\ w_2(t) \end{bmatrix}$ is the noise input vector,

where

$$A_{11} = \begin{bmatrix} -\frac{c_1}{m_1} & 0 & 0 & 0 & 0 & 0 & 0 & \frac{c_1}{m_1} & 0 & 0 \\ 0 & -\frac{c_2}{m_2} & 0 & 0 & 0 & 0 & 0 & 0 & \frac{c_2}{m_2} & 0 \\ 0 & 0 & -\frac{c_3+c_4}{m_3} & \frac{c_4}{m_3} & 0 & 0 & 0 & 0 & 0 & \frac{c_3}{m_3} \\ 0 & 0 & \frac{c_4}{m_4} & -\frac{c_4+c_5}{m_4} & \frac{c_5}{m_4} & 0 & 0 & 0 & 0 & 0 \\ 0 & 0 & 0 & \frac{c_5}{m_5} & -\frac{c_5+c_6}{m_5} & \frac{c_6}{m_5} & 0 & 0 & 0 & 0 \\ 0 & 0 & 0 & 0 & \frac{c_6}{m_6} & -\frac{c_6+c_7}{m_6} & \frac{c_7}{m_6} & 0 & 0 & 0 \\ 0 & 0 & 0 & 0 & 0 & \frac{c_7}{m_7} & -\frac{c_7}{m_7} & 0 & 0 & 0 \\ \alpha_1 c_1 & \alpha_4 c_2 & \alpha_5 c_3 & 0 & 0 & 0 & 0 & -\alpha_1 c_1 & -\alpha_4 c_2 & -\alpha_5 c_3 \\ \alpha_4 c_1 & \alpha_2 c_2 & \alpha_6 c_3 & 0 & 0 & 0 & 0 & -\alpha_4 c_1 & -\alpha_2 c_2 & -\alpha_6 c_3 \\ \alpha_5 c_1 & \alpha_6 c_2 & \alpha_3 c_3 & 0 & 0 & 0 & 0 & -\alpha_5 c_1 & -\alpha_6 c_2 & -\alpha_3 c_3 \end{bmatrix},$$

$$A_{12} = \begin{bmatrix} -\frac{k_1+k_{t1}}{m_1} & 0 & 0 & 0 & 0 & 0 & 0 & \frac{k_1}{m_1} & 0 & 0 \\ 0 & -\frac{k_2+k_{t2}}{m_2} & 0 & 0 & 0 & 0 & 0 & 0 & \frac{k_2}{m_2} & 0 \\ 0 & 0 & -\frac{k_3+k_4}{m_3} & \frac{k_4}{m_3} & 0 & 0 & 0 & 0 & 0 & \frac{k_3}{m_3} \\ 0 & 0 & \frac{k_4}{m_4} & -\frac{k_4+k_5}{m_4} & \frac{k_5}{m_4} & 0 & 0 & 0 & 0 & 0 \\ 0 & 0 & 0 & \frac{k_5}{m_5} & -\frac{k_5+k_6}{m_5} & \frac{k_6}{m_5} & 0 & 0 & 0 & 0 \\ 0 & 0 & 0 & 0 & \frac{k_6}{m_6} & -\frac{k_6+k_7}{m_6} & \frac{k_7}{m_6} & 0 & 0 & 0 \\ 0 & 0 & 0 & 0 & 0 & \frac{k_7}{m_7} & -\frac{k_7}{m_7} & 0 & 0 & 0 \\ \alpha_1 k_1 & \alpha_4 k_2 & \alpha_5 k_3 & 0 & 0 & 0 & 0 & -\alpha_1 k_1 & -\alpha_4 k_2 & -\alpha_5 k_3 \\ \alpha_4 k_1 & \alpha_2 k_2 & \alpha_6 k_3 & 0 & 0 & 0 & 0 & -\alpha_4 k_1 & -\alpha_2 k_2 & -\alpha_6 k_3 \\ \alpha_5 k_1 & \alpha_6 k_2 & \alpha_3 k_3 & 0 & 0 & 0 & 0 & -\alpha_5 k_1 & -\alpha_6 k_2 & -\alpha_3 k_3 \end{bmatrix},$$

$$A_{13} = \begin{bmatrix} \frac{k_{t1}}{m_1} & 0 & 0 & 0 & 0 & 0 & 0 & 0 & 0 & 0 \\ 0 & \frac{k_{t2}}{m_2} & 0 & 0 & 0 & 0 & 0 & 0 & 0 & 0 \end{bmatrix}^T, \quad A_{33} = \begin{bmatrix} -2\pi f_0 & 0 \\ 0 & -2\pi f_0 \end{bmatrix},$$

$$A_{21} = E_{10\times10}, \ A_{22} = 0_{10\times10}, \ A_{23} = 0_{10\times2}, \ A_{31} = 0_{2\times10}, \ A_{32} = 0_{2\times10}.$$

When the car is driving, the front and rear tires of the car will drive on the same road, the time interval is

$$\Delta t = \frac{a+b}{v}. \tag{37}$$

The Laplace transfer function between the road surface inputs at the front and rear tires and its second-order Pade approximation are expressed as

$$\frac{w_2(s)}{w_1(s)} = e^{-\Delta t s} = \frac{a_0 - a_1 s + a_2 s^2}{a_0 + a_1 s + a_2 s^2}, \tag{38}$$

where $a_0 = 12/\Delta t^2$, $a_1 = 6/\Delta t$, $a_2 = 1$, and $s$ is the Laplace operator.

Let the 2-dimensional additional state vector be $\mu = \begin{bmatrix} \mu_1 & \mu_2 \end{bmatrix}^T$, and transform Equation (35) into the space state equation as

$$\begin{cases} \dot{\mu}(t) = A_\mu \mu(t) + B_\mu w_1(t) \\ w_2(t) = w_1(t - \triangle t) = \mu_1(t) + w_1(t) \end{cases}, \tag{39}$$

where

$$A_\mu = \begin{bmatrix} 0 & 1 \\ -a_0 & -a_1 \end{bmatrix}, B_\mu = \begin{bmatrix} -2a_1 \\ 6a_0 \end{bmatrix}$$

Combining Equations (33) and (36), the suspension space state equation with additional state vector is obtained,

$$\begin{bmatrix} \dot{X} \\ \dot{\mu} \end{bmatrix} = \begin{bmatrix} A & GC_\mu \\ 0_{2\times2} & A_\mu \end{bmatrix} \begin{bmatrix} X \\ \mu \end{bmatrix} + \begin{bmatrix} B \\ 0_{2\times2} \end{bmatrix} U + \begin{bmatrix} GE_\mu \\ B_\mu \end{bmatrix} w_1. \tag{40}$$

where

$$C_\mu = \begin{bmatrix} 0 & 0 \\ 1 & 0 \end{bmatrix}, E_\mu = \begin{bmatrix} 1 \\ 1 \end{bmatrix}.$$

Let

$$X_P = \begin{bmatrix} X \\ \mu \end{bmatrix}, A_P = \begin{bmatrix} A & GC_\mu \\ 0_{2\times2} & A_\mu \end{bmatrix}, B_P = \begin{bmatrix} B \\ 0_{2\times2} \end{bmatrix}$$

and

$$G_P = \begin{bmatrix} GE_\mu \\ B_\mu \end{bmatrix},$$

then Equation (40) is, in which the state vector has 24 numbers.

$$\dot{X}_P = A_P X_P + B_P U + G_P w_1. \tag{41}$$

### 3. $L_2$ State Gain Feedback Control Algorithm for Active Suspension

*3.1. $L_2$ State Gain Feedback Control*

The non-homogeneous state equation of the closed-loop linear system is defined as

$$\dot{X}_P(t) = A_P X_P(t) + B_P U(t) + G_P w_1(t), \tag{42}$$

$$Y(t) = C X_P(t) + D U(t). \tag{43}$$

Let the feedback control input be $U(t) = K X_P(t)$, and substitute it into Equations (42) and (43),

$$\dot{X}_P(t) = (A_P + B_P K) X_P(t) + G_P w_1(t), \tag{44}$$

$$Y(t) = (C + DK) X_P(t). \tag{45}$$

According to the principle of $L_2$ state gain feedback control [30–32], the $L_2$ gain of the closed-loop system should be less than any positive number $\gamma$,

$$\frac{\|Y(t)\|}{\|w_1(t)\|} < \gamma. (\gamma > 0) \tag{46}$$

Therefore, $\gamma$ is the upper bound of the $L_2$ gain of the closed-loop system. When the above condition is satisfied, in order to ensure the stability of the feedback system, it needs to be met,

$$\dot{V}[X_P(t)] + \gamma^{-1} Y^T(t) Y(t) - \gamma w_1^T(t) w_1(t) < 0, \tag{47}$$

where $\dot{V}[X_P(t)]$ is the second-order Lyapunov function, and its expression is

$$\dot{V}[X_P(t)] = X_P^T(t) P X_P(t), \tag{48}$$

where $P \succ 0$ is a symmetric positive definite matrix. Substituting Equations (44), (45), and (48) into the linear matrix inequality (47),

$$X_P^T(t)(A_P + B_P K)^T P X_P(t) + w_1^T(t) G_P^T P X_P(t) + X_P^T(t) P (A_P + B_P K) X_P(t) + X_P^T(t) P G_P w_1(t)$$
$$+ \gamma^{-1} X_P^T(t)(C + DK)^T(C + DK) X_P(t) - \gamma w_1^T(t) w_1(t) < 0 \tag{49}$$

Let $\xi(t) = \begin{bmatrix} X_P(t) & w_1(t) \end{bmatrix}$, then Equation (49) can be expressed as

$$\xi(t) \beta \xi^T(t) < 0, \tag{50}$$

where $\beta = \begin{bmatrix} (A_P + B_P K)^T P + P(A_P + B_P K) + \gamma^{-1}(C + DK)^T(C + DK) & P G_P \\ G_P^T P & -\gamma I \end{bmatrix}$.

Suppose the partition matrix $M = \begin{bmatrix} M_{11} & M_{12} \\ M_{21} & M_{22} \end{bmatrix}$, where $M_{22}$ is non-zero value, then the Schur complement matrix of the partition matrix $M$ with respect to $M_{22}$ is

$$M/M_{22} = M_{11} - M_{12} M_{22}^{-1} M_{21}. \tag{51}$$

Since the matrix $\beta$ satisfies the form of Equation (51), namely,

$$\beta = \begin{bmatrix} (A_P + B_P K)^T P + P(A_P + B_P K) & P G_P \\ G_P^T P & -\gamma I \end{bmatrix} - \begin{bmatrix} (C + DK)^T(-\gamma^{-1})(C + DK) & 0 \\ 0 & 0 \end{bmatrix}. \tag{52}$$

Then, the matrix $\beta$ is rewritten as a Schur complement matrix and substituted into Equation (50),

$$M = \begin{bmatrix} (A_P + B_P K)_T P + P(A_P + B_P K) & P G_P & (C + DK)^T \\ G_P^T P & -\gamma I & 0 \\ C + DK & 0 & -\gamma I \end{bmatrix} \prec 0, \tag{53}$$

where $M \prec 0$ indicates that the partition matrix $M$ is a negative definite matrix.

Multiply both sides of the matrix $M$ by the diagonal matrix $diag(P^{-1}, I, I)$ respectively, and let $Z = P^{-1}$ and $W = KZ$ to get

$$
\begin{bmatrix}
ZA_P^T + W^T B_P^T + A_P Z + BW & G & Z^T C^T + W^T D^T \\
G_P^T & -\gamma I & 0 \\
CZ + DW & 0 & -\gamma I
\end{bmatrix} \prec 0. \tag{54}
$$

Therefore, according to the analysis in this section, $\gamma$ is the upper bound of the state feedback gain of the system at all times. The magnitude of feedback control force is closely related to $\gamma$. In order to make the system have a good control effect, the problem of solving the $L_2$ state gain can be transformed into a linear matrix inequality problem of solving the minimum value of $\gamma$.

### 3.2. $L_2$ State Gain of the Driver-Active Suspension Model

In order to improve the ride comfort, according to the international standard file ISO 2631, take the acceleration of the head and neck, the acceleration of the hip and thigh (that is, the seat support surface), the pitch angle acceleration, the body acceleration of the front axle, the suspension working space of the front axle, the dynamic tire deflection of the front tire, the body acceleration of the rear axle, the suspension working space of the rear axle and the dynamic tire deflection of the rear tire as performance indexes, that is,

$$
Y = [\rho_1 \ \ddot{x}_7 \quad \rho_2 \ddot{x}_4 \quad \rho_3 \ddot{\theta} \quad \rho_4 \ddot{x}_8 \quad \rho_5(x_8 - x_1) \quad \rho_6(x_1 - x_{01}) \quad \rho_7 \ddot{x}_9 \quad \rho_8(x_9 - x_2) \quad \rho_9(x_2 - x_{02}) \ ]^T, \tag{55}
$$

where $\rho = diag(\rho_1, \rho_2, \rho_3, \rho_4, \rho_5, \rho_6, \rho_7, \rho_8, \rho_9)$ is the weighting coefficient matrix of each performance index, and its value is the tendency of control rule to each performance index.

According to the performance index variables, the state index equation of the system is established as follows:

$$
Y = CX_P + DU, \tag{56}
$$

where $C = \begin{bmatrix} C_1 & C_2 & C_3 \end{bmatrix}$ is the output state coefficient matrix, $D$ is the input–output coupling matrix,

$$
C_1 = \begin{bmatrix}
0 & 0 & \rho_1 & 0 & 0 & \frac{\rho_1 c_7}{m_7} & -\frac{\rho_1 c_7}{m_7} & 0 & 0 & 0 \\
0 & 0 & \frac{\rho_2 c_4}{m_4} & -\frac{\rho_2(c_4+c_5)}{m_4} & \frac{\rho_2 c_5}{m_4} & 0 & 0 & 0 & 0 & 0 \\
-\frac{\rho_3 a c_1}{I} & \frac{\rho_3 b c_2}{I} & -\frac{\rho_3 c c_3}{I} & 0 & 0 & 0 & 0 & \frac{\rho_3 a c_1}{I} & -\frac{\rho_3 b c_2}{I} & \frac{\rho_3 c c_3}{I} \\
\rho_4 \alpha_1 c_1 & \rho_4 \alpha_4 c_2 & \rho_4 \alpha_5 c_3 & 0 & 0 & 0 & 0 & -\rho_4 \alpha_1 c_1 & -\rho_4 \alpha_4 c_2 & -\rho_4 \alpha_5 c_3 \\
0 & 0 & 0 & 0 & 0 & 0 & 0 & 0 & 0 & 0 \\
0 & 0 & 0 & 0 & 0 & 0 & 0 & 0 & 0 & 0 \\
\rho_7 \alpha_4 c_1 & \rho_7 \alpha_2 c_2 & \rho_7 \alpha_6 c_3 & 0 & 0 & 0 & 0 & -\rho_7 \alpha_4 c_1 & -\rho_7 \alpha_2 c_2 & -\rho_7 \alpha_6 c_3 \\
0 & 0 & 0 & 0 & 0 & 0 & 0 & 0 & 0 & 0 \\
0 & 0 & 0 & 0 & 0 & 0 & 0 & 0 & 0 & 0
\end{bmatrix},
$$

$$
C_2 = \begin{bmatrix}
0 & 0 & \rho_1 & 0 & 0 & \frac{\rho_1 k_7}{m_7} & -\frac{\rho_1 k_7}{m_7} & 0 & 0 & 0 \\
0 & 0 & \frac{\rho_2 k_4}{m_4} & -\frac{\rho_2(k_4+k_5)}{m_4} & \frac{\rho_2 k_5}{m_4} & 0 & 0 & 0 & 0 & 0 \\
-\frac{\rho_3 a k_1}{I} & \frac{\rho_3 b k_2}{I} & -\frac{\rho_3 c k_3}{I} & 0 & 0 & 0 & 0 & \frac{\rho_3 a k_1}{I} & -\frac{\rho_3 b k_2}{I} & \frac{\rho_3 c k_3}{I} \\
\rho_4 \alpha_1 k_1 & \rho_4 \alpha_4 k_2 & \rho_4 \alpha_5 k_3 & 0 & 0 & 0 & 0 & -\rho_4 \alpha_1 k_1 & -\rho_4 \alpha_4 k_2 & -\rho_4 \alpha_5 k_3 \\
-\rho_5 & 0 & 0 & 0 & 0 & 0 & 0 & \rho_5 & 0 & 0 \\
\rho_6 & 0 & 0 & 0 & 0 & 0 & 0 & 0 & 0 & 0 \\
\rho_7 \alpha_4 k_1 & \rho_7 \alpha_2 k_2 & \rho_7 \alpha_6 k_3 & 0 & 0 & 0 & 0 & -\rho_7 \alpha_4 k_1 & -\rho_7 \alpha_2 k_2 & -\rho_7 \alpha_6 k_3 \\
0 & -\rho_8 & 0 & 0 & 0 & 0 & 0 & 0 & \rho_8 & 0 \\
0 & \rho_9 & 0 & 0 & 0 & 0 & 0 & 0 & 0 & 0
\end{bmatrix},
$$

$$
C_3 = \begin{bmatrix}
0 & 0 & 0 & 0 & 0 & 0 & 0 & 0 & -\rho_9 \\
0 & 0 & 0 & 0 & 0 & -\rho_6 & 0 & 0 & 0 \\
0 & 0 & 0 & 0 & 0 & 0 & 0 & 0 & 0 \\
0 & 0 & 0 & 0 & 0 & 0 & 0 & 0 & 0
\end{bmatrix}^T, \quad D = \begin{bmatrix}
0 & 0 & -\frac{\rho_3 a}{I} & \rho_4 \alpha_1 & 0 & 0 & \rho_7 \alpha_4 & 0 & 0 \\
0 & 0 & \frac{\rho_3 b}{I} & \rho_4 \alpha_4 & 0 & 0 & \rho_7 \alpha_2 & 0 & 0
\end{bmatrix}^T.
$$

According to the above $L_2$ state gain feedback control algorithm, taking the Ford Granada as an example, the driver's active suspension model is established. The parameters of the established active suspension system model are shown in Table 1, and the parameters of the established driver model are shown in Table 2.

**Table 1.** Model parameters of semi-vehicle active suspension.

| Model Parameter | Value | Model Parameter | Value | Model Parameter | Value |
|---|---|---|---|---|---|
| $m/\text{kg}$ | 690 | $I/(\text{kg}\cdot\text{m}^2)$ | 1222 | $m_1/\text{kg}$ | 40 |
| $m_2/\text{kg}$ | 45 | $k_1/(\text{N}\cdot\text{m}^{-1})$ | 17,000 | $k_2/(\text{N}\cdot\text{m}^{-1})$ | 22,000 |
| $k_{t1}/(\text{N}\cdot\text{m}^{-1})$ | 192,000 | $k_{t2}/(\text{N}\cdot\text{m}^{-1})$ | 192,000 | $c_1/(\text{N}\cdot\text{s}\cdot\text{m}^{-1})$ | 1500 |
| $c_2/(\text{N}\cdot\text{s}\cdot\text{m}^{-1})$ | 1500 | $f/\text{Hz}$ | 0.1 | $n_0$ | 0.1 |
| $a/\text{m}$ | 1.011 | $b/\text{m}$ | 1.803 | $c/\text{m}$ | 0.3 |

**Table 2.** Model parameters of driver model.

| Model Parameter | Value | Model Parameter | Value | Model Parameter | Value |
|---|---|---|---|---|---|
| $m_3/\text{kg}$ | 15 | $k_3/(\text{N}\cdot\text{m}^{-1})$ | 31,000 | $c_3/(\text{N}\cdot\text{s}\cdot\text{m}^{-1})$ | 830 |
| $m_4/\text{kg}$ | 12.78 | $k_4/(\text{N}\cdot\text{m}^{-1})$ | 90,000 | $c_4/(\text{N}\cdot\text{s}\cdot\text{m}^{-1})$ | 2064 |
| $m_5/\text{kg}$ | 8.62 | $k_5/(\text{N}\cdot\text{m}^{-1})$ | 162,800 | $c_5/(\text{N}\cdot\text{s}\cdot\text{m}^{-1})$ | 4585 |
| $m_6/\text{kg}$ | 28.49 | $k_6/(\text{N}\cdot\text{m}^{-1})$ | 183,000 | $c_6/(\text{N}\cdot\text{s}\cdot\text{m}^{-1})$ | 4750 |
| $m_7/\text{kg}$ | 5.31 | $k_7/(\text{N}\cdot\text{m}^{-1})$ | 310,000 | $c_7/(\text{N}\cdot\text{s}\cdot\text{m}^{-1})$ | 400 |

The performance index of vehicle suspension system will change with the change of weighting coefficient. Therefore, the determination of the weighting coefficient determines the control effect of the controller. In this paper, genetic algorithm is used to solve the weighting coefficient by referring to references [33,34]. The algorithm flow is shown in Figure 3.

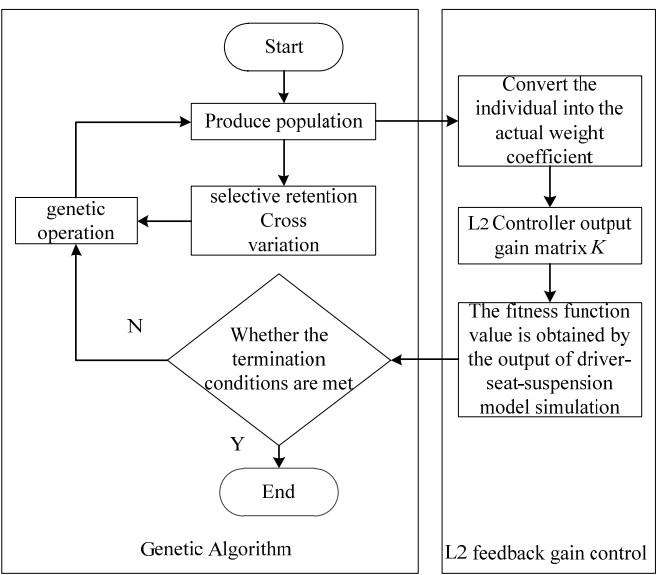

**Figure 3.** Schematic diagram of genetic algorithm.

The order of magnitude and the unit of the nine performance indexes of the driver seat-active suspension is different. For normalization comparison, the fitness function of genetic algorithm is set to

$$\min f(X) = \sum_{i=1}^{9} \frac{RMS[\triangle_i(\rho)]}{RMS[\triangle_{ip}(\rho)]}. \tag{57}$$

The constraint condition is

$$RMS[\triangle_i(\rho)] < RMS[\triangle_{ip}(\rho)], \tag{58}$$

where *RMS* is the root mean square value of the 9 performance indicators of the system, $\Delta_i(\rho)$ is the performance index of active suspension, $\Delta_{iP}(\rho)$ is the performance index of passive suspension, $\rho = [\rho_1, \rho_2, \rho_3, \rho_4, \rho_5, \rho_6, \rho_7, \rho_8, \rho_9]$ is the weighting coefficient matrix of each performance index.

Set the initial search range of the genetic algorithm. The range of $\rho_1$ is [1, 1], the range of $\rho_2$ is [1, 1000], the range of $\rho_3$ is [1, 1000], the range of $\rho_4$ is [1, 1000], the range of $\rho_5$ is [1000, 10,000], the range of $\rho_6$ is [1000, 10,000], the range of $\rho_7$ is [1, 1000], the range of $\rho_8$ is [10,000, 50,000] and the range of $\rho_9$ is [10,000, 50,000].

The weighting coefficients of the performance index of the $L_2$ state gain feedback control in this paper are obtained in Figure 4.

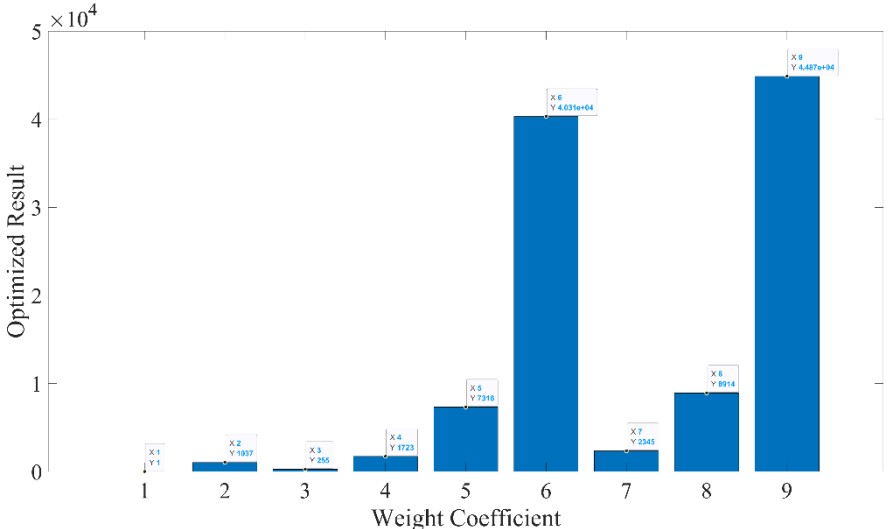

**Figure 4.** Optimization results of genetic algorithm.

The weighting coefficients $\rho = [1, 1037, 255, 1723, 7316, 40312, 2345, 8914, 44872]$. According to the model parameters established in this paper, through solving the linear matrix inequality of the driver-active suspension model, the minimum value of the upper bound $\gamma$ of $L_2$ gain of the closed-loop system satisfying the inequality (54) is 13.686. The corresponding $L_2$ state gain feedback matrix is

$$K = \begin{bmatrix} K_1 & K_2 \end{bmatrix}^T, \tag{59}$$

where

$$K_1 = \begin{bmatrix} -1136.2 & -1983.9 & -7534.4 & 5934.1 & 1324.9 & 3.4 & -521.3 & -1396.14 & -12.5 & -1583.4 & -1684.4 & \cdots \\ -3355.6 & -6103.0 & 6844.6 & 1324.8 & 5.6 & -488.8 & -198.4 & 199.2 & -241.6 & -4442.5 & -96.3 & 24.1 & 11.8 \end{bmatrix}^T,$$
$$K_2 = \begin{bmatrix} -483.1 & -3587.4 & -6019.9 & 6357.5 & 2272.2 & -469.1 & -55.4 & -247.9 & 88.7 & -67.1 & -486.5 & \cdots \\ -5541.2 & -4876.8 & 6411.7 & 2069.6 & 19.4 & -99.4 & -173.5 & 329.9 & -31.7 & 180.7 & -102.8 & -28.1 & 59.7 \end{bmatrix}^T.$$

### 3.3. Energy Consumption of Multi-Link Active Suspension

The structure diagram of the multi-link active suspension is shown in Figure 5.

As shown in Figure 5a,b, the relationship between the axial control force of the real suspension actuator and the vertical control force in the simplified model is

$$u_y = u \cos \alpha. \tag{60}$$

Then, the real active control force of the actuator is

$$u = \frac{u_y}{\cos \alpha}. \tag{61}$$

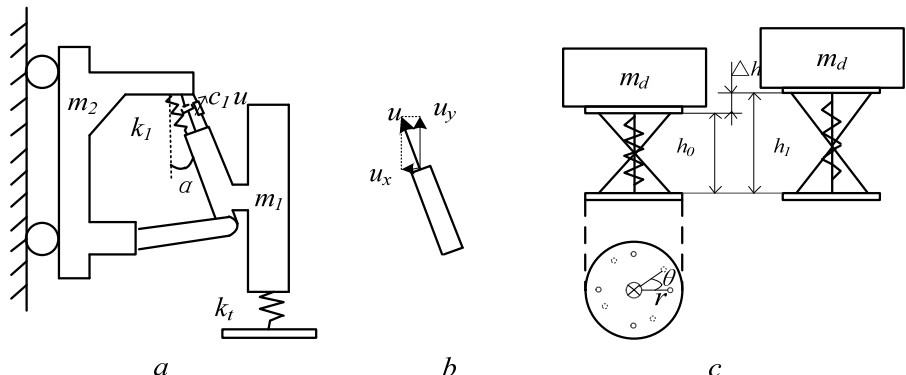

**Figure 5.** The structure diagram of the multi-link active suspension. (**a**) Real model of suspension. (**b**) Force analysis. (**c**) Kinematic analysis of multi-link suspension.

The relationship between the height of the suspension and the rotation angle of the lower platform is

$$h = \sqrt{l^2 - 4r^2 \sin^2 \frac{\theta}{2}}, \tag{62}$$

where $h$ is the suspension height, $l$ is the length of the link, that is, the original height of the suspension, $r$ is the distance from the center of the link to the center of the platform and $\theta$ is the angle of rotation of the platform relative to the original position.

For a single active suspension, according to the law of conservation of energy, it can be known that

$$u_y(h_1 - h_0) = M(\theta_1 - \theta_0) = m_d(h_1 - h_0) + \frac{1}{2}nm_l(h_1 - h_0) - \frac{1}{2}k[(l - h_0)^2 - (l - h_1)^2]^2, \tag{63}$$

where $u_y$ is the vertical equivalent active control force, $h_0$ is the suspension height without active control force, $h_1$ is the suspension height with active control force, $M$ is the torque of the lower platform, $\theta_0$ is the angle of rotation of the lower platform without active control force, $\theta_1$ is the angle of rotation of the lower platform with active control force, $m_d$ is the weight of the body and the upper platform of the suspension, $n = 4$ is the number of link, and $m_l$ is the weight of a single link.

The circuit diagram of the active suspension actuator is shown in Figure 6, where $M$ is the permanent magnet brushless DC, $L$ is the inductor, $U$ is the supply voltage, $I$ is the armature circuit current, and $R$ is the armature circuit internal resistance.

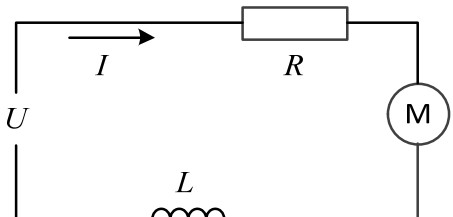

**Figure 6.** Control circuit diagram of multi-link active suspension.

The relationship between the torque required by the suspension lower platform and the motor output torque is

$$M = K_T M_0 = \eta_B i M_0, \tag{64}$$

where $K_T$ is the torque transfer coefficient, $M_0$ is the motor output torque, $\eta_B$ is the transmission efficiency of the mechanism, and $i$ is the transmission ratio of the growth mechanism.

The parameter nature of the actuator is

$$U_A = K_A x_A, \tag{65}$$

$$F_U = K_A I, \tag{66}$$

where $U_A$ is the electromotive force of the motor, $x_A$ is the axial speed of suspension, $K_A$ is the suspension conversion to drive motor constant, $F_U$ is active control force.

$$K_A = K_T K_M, \tag{67}$$

where $K_M$ is the motor torque constant.

The motor equations are

$$\begin{cases} U_m = \frac{nP}{60a} \Phi n_m \\ \Phi = B_A S = B_A D_A l_A \\ P_m = \frac{M_0 n_m}{9550} \end{cases}, \tag{68}$$

where $U_m$ is the rated power of the motor, $n$ is the number of electric drive conductors, $P$ is the pole log of the motor, $a$ is the logarithm of the parallel branches, $\Phi$ is the magnetic flux passing through the coil, $n_m$ is rated motor speed, $D_A$ is the outer diameter, $B_A$ is air gap magnetic density, $l_A = \mu D_A$ is the electric drive length, and $P_m$ is rated power of the motor.

The relationship between active control force $F_U$ and current $I$ is

$$F_U = \frac{955 n i D_A B_A l_A P I}{18 \eta_B r_A a}, \tag{69}$$

where $\mu$ is the ratio of electric drive length to outside diameter and $r_A$ is the radius of the rotating platform.

$$D_A = \lambda_A \left( \frac{6.1 \times 10^8 \times P_S}{\delta A B_A n_m \mu K_P} \right)^{\frac{1}{3}}, \tag{70}$$

where $\lambda_A$ is the outer diameter correction factor, $P_S$ is the apparent power, $\delta$ is the polar arc coefficient, $A$ is electrical load, and $K_P$ is the short moment coefficient.

The real parameters of the multi-link active suspension motor are shown in Table 3.

**Table 3.** The parameters of motor.

| Parameter | Value | Parameter | Value | Parameter | Value | Parameter | Value |
|---|---|---|---|---|---|---|---|
| $n_m/(r/\text{min})$ | 2300 | $R/\Omega$ | 10 | $B_A/\text{Gs}$ | 3200 | $P_S/\text{W}$ | 67.78 |
| $r_A/\text{m}$ | 0.075 | $A/(A/\text{cm})$ | 80 | $P_m/\text{kW}$ | 1.35 | $\delta$ | 0.72 |
| $n$ | 432 | $\lambda_A$ | 0.556 | $\mu$ | 1.125 | $i$ | 6 |
| $P$ | 2 | $K_P$ | 0.951 | $a$ | 2 | $\eta_B$ | 0.9 |

According to the motor and structural parameters, $K_A = 98.27(\text{N/A})$ is calculated.

According to the energy consumption calculation method [35,36], for the model established in this paper, the total energy consumption power of the multi-link active suspension $E$ is

$$E = E_f + E_r = F_f(\dot{x}_8 - \dot{x}_1) + F_r(\dot{x}_9 - \dot{x}_2), \tag{71}$$

where $E_f$ is the power consumed by the front axle active suspension, $E_r$ is the power consumed by the rear axle active suspension, $F_f$ is the active control force of the front axle suspension, and $F_r$ is the active control force of the rear axle suspension.

Then the energy consumption power used by the active element to control vibration per each unit of mass is

$$E_u = \frac{E_f + E_r}{m + m_d} = \frac{E_f + E_r}{m + m_3 + m_4 + m_5 + m_6 + m_7}. \tag{72}$$

### 3.4. Simulation and Experiment of Active Suspension

According to the above $L_2$ state gain feedback control algorithm and model parameters, a 9-DOF driver seat-active suspension model, a driver seat-passive suspension model and a traditional LQG semi-vehicle active suspension model that do not consider the driver are established, respectively. The suspension model is simulated by using the class B road conditions and 20 m/s vehicle speed conditions in the international standardization document ISO/TC 108/SC2N67. The active control force and energy consumption power of front axle and rear axle actuators based on $L_2$ control driver model and traditional LQG algorithm [18] are shown in Figure 7.

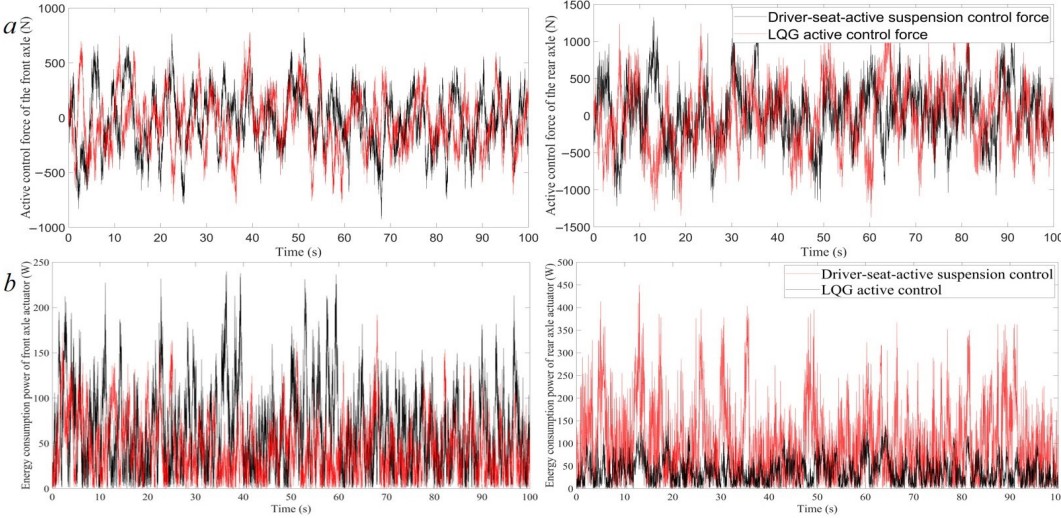

**Figure 7.** Comparison of actuator performance. (**a**) Active control force of the actuator. (**b**) Energy consumption power of the actuator.

According to Figure 7a curve, the RMS values of the active control force of the front axle and rear axle actuators of the driver seat-active suspension are 265.79 N and 390.68 N, respectively. The RMS values of the active control force of the LQG active suspension are 255.02 N and 429.81 N, respectively.

It can be seen from Figure 7b, the mean value of the energy consumption power of the front axle actuator of the driver seat-active suspension is 77.6034 W, the mean value of the LQG front axle actuator is 55.1376 W, the mean value of the energy consumption power of the rear axle actuator of the driver seat-active suspension is 47.2937 W and the mean value of the LQG rear axle actuator is 132.3449 W. The new control algorithm can decrease the total energy cost from 180 W to 133 W. According to Equation (70), the mean value of power consumed per unit of sprung mass is 0.1523 W/Kg less than 0.2286 W/Kg of LQG algorithm. Compared with the LQG control algorithm, the algorithm established in this paper has a better energy consumption.

According to Equations (63) and (69), the drive current response curve of the active suspension drive motor is obtained, and as shown in Figure 8.

It can be seen from Figure 8, the driving current required by the front and rear axle actuators is less than 15 A, within the rated current range. A positive current means that the direction of the active control force is upward, and vice versa. By calculation, the RMS value of the driving current of the front axle actuator is 2.5951 A. The RMS value of the driving current of the rear axle actuator is 4.3737 A.

The simulation results of the semi-vehicle 9-DOF driver seat-active suspension model, the LQG semi-vehicle-active suspension model without considering the driver and the driver seat-passive suspension model are shown in Figure 9. The RMS values and the absolute value of the mean value of the performance indexes of the three different models are calculated respectively, and as shown in Table 4. After the energy supplement system

has been designed, the control efficiency could be calculated using the ratio between the currents or powers.

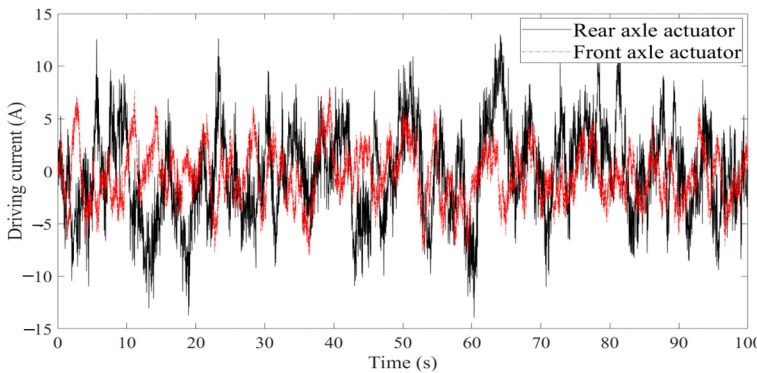

**Figure 8.** Drive current of active suspension actuators.

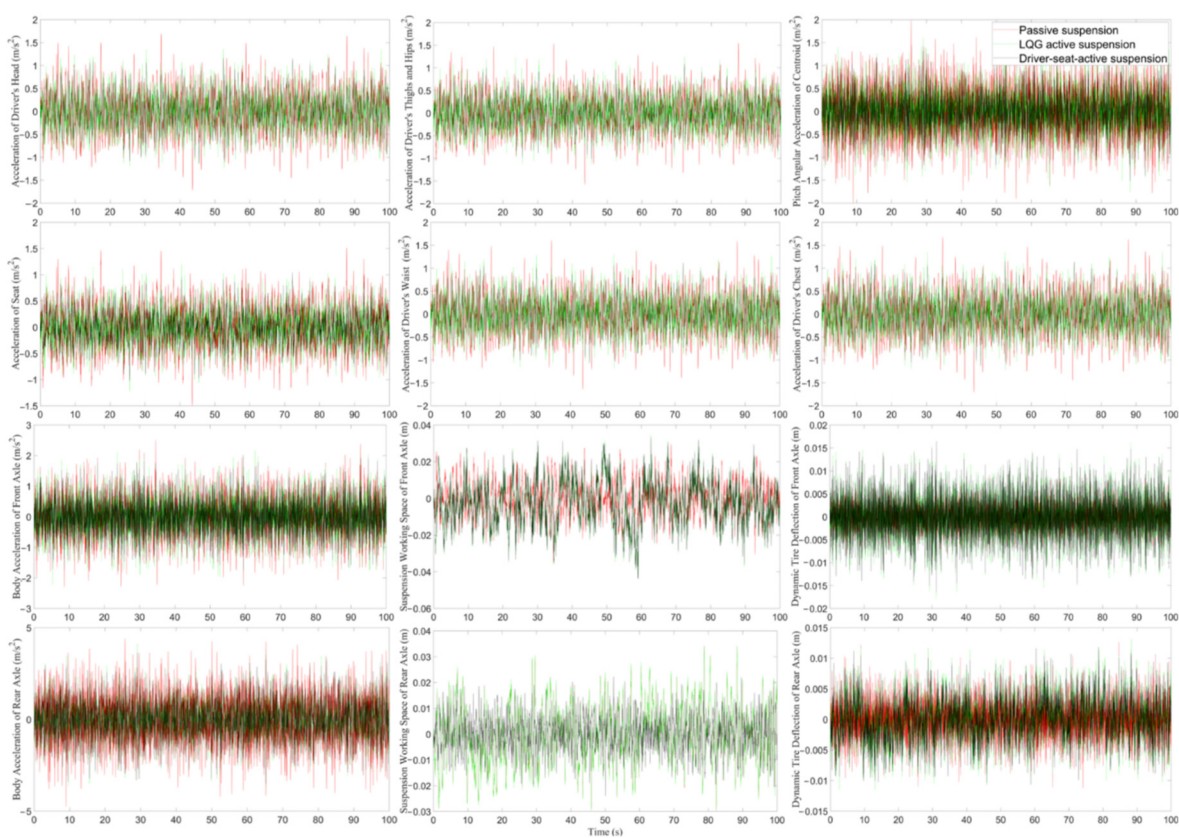

**Figure 9.** Comparison of suspension performance indexes.

It can be seen from Figure 9 and Table 4 that the system model and control algorithm established in this paper can significantly improve the ride comfort of the driver. Compared with the passive suspension and the LQG active suspension without considering the driver, the RMS value of the acceleration on the driver's head is respectively reduced by 27.5% and 10.9%. The RMS value of the acceleration on the driver's hip and thigh (that is, the seat support surface) is, respectively, reduced by 29.9% and 15.9%. The RMS value of the pitch angle acceleration experienced by the driver is reduced by 27.2% and 6.4%, respectively. The RMS value of the acceleration on the driver's waist is respectively reduced by 28.7% and 6.7%. The RMS value of the acceleration on the driver's chest is respectively reduced by 27.7% and 7.5%. Compared with the passive suspension, the absolute value of the mean value of these five performance indexes decreased by 94.5%, 89.5%, 26.9%, 89.2%, and

90.1%, respectively. Compared with the LQG active suspension without considering the driver, the absolute value of the mean value of these five performance indexes respectively decreased by 50.0%, 20.0%, 38.7%, 58.8%, and 61.9%.

**Table 4.** Comparison of root mean square values of suspension performance indexes.

| Performance Index | Unit | Driver Seat-Active Suspension | | Driver Seat-Passive Suspension | | LQG-Active Suspension | |
|---|---|---|---|---|---|---|---|
| | | RMS | Mean | RMS | Mean | RMS | Mean |
| Acceleration of the driver's head and neck $(\ddot{x}_7)$ | m/s$^2$ | 0.3371 | 0.00002 | 0.4651 | 0.00037 | 0.3782 | 0.00004 |
| Acceleration of the driver's hip and thigh $(\ddot{x}_4)$ | m/s$^2$ | 0.2989 | 0.00004 | 0.4264 | 0.00038 | 0.3557 | 0.00005 |
| Pitch angle acceleration $(\ddot{\theta})$ | m/s$^2$ | 0.3693 | 0.00019 | 0.5073 | 0.00026 | 0.3944 | 0.00031 |
| Acceleration of the seat $(\ddot{x}_3)$ | m/s$^2$ | 0.2778 | 0.00005 | 0.4029 | 0.00051 | 0.3148 | 0.00007 |
| Acceleration of the driver's waist $(\ddot{x}_5)$ | m/s$^2$ | 0.3169 | 0.00007 | 0.4447 | 0.00065 | 0.3398 | 0.00017 |
| Acceleration of the driver's chest $(\ddot{x}_6)$ | m/s$^2$ | 0.3340 | 0.00008 | 0.4619 | 0.00081 | 0.3612 | 0.00021 |
| Body acceleration of the front axle $(\ddot{x}_8)$ | m/s$^2$ | 0.5216 | 0.00028 | 0.6466 | 0.00032 | 0.5284 | 0.00029 |
| Suspension working space of the front axle $(x_8 - x_1)$ | mm | 9.1 | 0.00599 | 9.1 | 0.00772 | 11.7 | 0.00624 |
| Dynamic tire deflection of the front axle $(x_1 - x_{01})$ | mm | 2.9 | 0.00009 | 3.1 | 0.00101 | 4.3 | 0.00011 |
| Body acceleration of the rear axle $(\ddot{x}_9)$ | m/s$^2$ | 0.9383 | 0.00098 | 1.2033 | 0.00106 | 0.9962 | 0.00101 |
| Suspension working space of the rear tire $(x_9 - x_2)$ | mm | 6.6 | 0.54406 | 6.7 | 0.00161 | 10.3 | 0.60027 |
| Dynamic tire deflection of the rear tire $(x_2 - x_{02})$ | mm | 2.9 | 0.12415 | 3.0 | 0.00094 | 3.3 | 0.1 1438 |

In addition, the $L_2$ state gain feedback control has an ideal control effect on suspension and passive components of the body. After the driver–seat–suspension model adopts the $L_2$ state gain feedback active control, compared with the passive suspension and the LQG active suspension without considering the driver, the RMS value of the body acceleration of the front axle is respectively reduced by 19.3% and 1.3%. The RMS value of the body acceleration of the rear axle is respectively reduced by 22.0% and 5.8%. The RMS value of the dynamic tire deflection of the front tire is respectively reduced by 6.5% and 32.6%. The RMS value of the dynamic tire deflection of the rear tire is respectively reduced by 3.3% and 12.1%. There is a coupling relationship between the body acceleration and the suspension working space. When the body acceleration is greatly increased, the suspension working space will deteriorate. According to the weighted factor optimization of $L_2$ control algorithm in Section 3.2, although the suspension working space of front axle and rear axle is not significantly optimized, compared with the LQG active suspension without considering the driver, the suspension working space of front axle and rear axle is respectively reduced by 22.2% and 35.9%.

In order to further verify the influence of the model and control algorithm established in this paper on driving comfort, the power spectral density function is obtained by spectrum analysis of acceleration of the driver's head and neck, acceleration of the driver's hip and thigh, and pitch angle acceleration. The power spectrum estimation of the performance index in the time domain is estimated by periodogram method.

$$[Pxx, \omega] = periodogram(x, [], N, f_x), \tag{73}$$

where $Pxx$ is the power spectral density of random sequence at the corresponding frequency $\omega$, $x$ is random sequence, $N$ is the number of random sequence data in the time domain, and $f_x$ is the sampling signal frequency. In this paper, the sampling time interval is 0.01 s , so $f_x = 100$. The power spectral density functions of the three performance indicators are shown in Figure 10.

It can be seen from Figure 10, the passive suspension and driver seat-active suspension have roughly the same waveforms. The power spectral density of acceleration on the driver's head and neck and acceleration on the driver's hip and thigh are all concentrated in the frequency range of 0 to 10 Hz, and there are two peaks within this frequency range. The power spectral density of the pitch angular acceleration mainly concentrates in the

frequency range of 0 to 20 Hz, and the peak value appears in the frequency range of 0 to 10 Hz and 10–20 Hz, respectively.

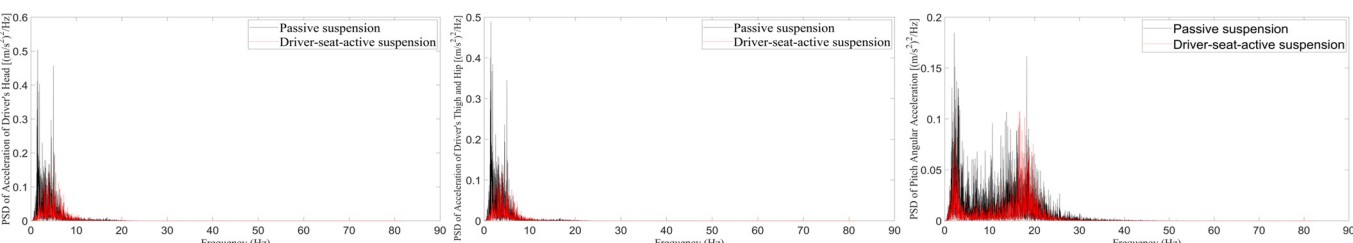

**Figure 10.** The power spectral density function of the performance indexes.

The power spectrum density function obtained above is frequency-weighted according to the international standard document ISO 2631, and the RMS value of the weighted acceleration of the driver seat-active suspension is $a_{WA} = 0.407$ m/s$^2$ and the RMS value of the weighted acceleration of the passive suspension is $a_{WP} = 0.628$ m/s$^2$, so that the driver is in a relatively comfortable range in the driver seat-active suspension system, and in a relatively uncomfortable range in the passive suspension system. Therefore, the driving comfort of the driver can be improved after the the $L_2$ state gain feedback control is adopted in vehicle suspension.

## 4. Adaptive Kalman Filter Algorithm for Driver Suspension Model

In the previous research on the driver-active suspension model, the control of the active suspension actuator is all based on the known input variables of the controller. However, in the process of vehicle driving, it is difficult to directly measure the performance indexes of the driver, such as the acceleration of the driver's head and neck, the acceleration of the driver's hip and thigh. Moreover, there will always be random noise interference during the measurement process, which leads to the unsatisfactory control effect of the active suspension actuator. Therefore, this paper uses the Kalman filter algorithm to optimally estimate the driver's state that cannot be directly measured and remove the measurement noise.

### 4.1. Discretization of Driver-Suspension State Equation

This paper chooses the convenient and reliable measurement of the body acceleration of the front axle, suspension working space of the front axle, dynamic tire deflection of the front tire, body acceleration of the rear axle, suspension working space of the rear axle, and dynamic tire deflection of the rear tire as observations, namely,

$$Z = \begin{bmatrix} \ddot{x}_8 & x_8 - x_1 & x_1 - x_{01} & \ddot{x}_9 & x_9 - x_2 & x_2 - x_{02} \end{bmatrix}^T. \tag{74}$$

According to the observations, the observation equation expression of the system is established as

$$Z = EX_P + FU, \tag{75}$$

where $E$ is the observation state coefficient matrix of system, $F$ is the input-observation coupling matrix,

$$E = \begin{bmatrix} \alpha_1 c_1 & \alpha_4 c_2 & \alpha_5 c_3 & 0 & 0 & 0 & 0 & -\alpha_1 c_1 & -\alpha_4 c_2 & -\alpha_5 c_3 & \alpha_1 k_1 & \alpha_4 k_2 & \alpha_5 k_3 & 0 & 0 & 0 & 0 & -\alpha_1 k_1 & -\alpha_4 k_2 & -\alpha_5 k_3 & 0 & 0 & 0 & 0 \\ 0 & 0 & 0 & 0 & 0 & 0 & 0 & 0 & 0 & 0 & -1 & 0 & 0 & 0 & 0 & 0 & 0 & 1 & 0 & 0 & 0 & 0 & 0 & 0 \\ 0 & 0 & 0 & 0 & 0 & 0 & 0 & 0 & 0 & 0 & 1 & 0 & 0 & 0 & 0 & 0 & 0 & 0 & 0 & 0 & -1 & 0 & 0 & 0 \\ \alpha_4 c_1 & \alpha_2 c_2 & \alpha_6 c_3 & 0 & 0 & 0 & 0 & -\alpha_4 c_1 & -\alpha_2 c_2 & -\alpha_6 c_3 & \alpha_4 k_1 & \alpha_2 k_2 & \alpha_6 k_3 & 0 & 0 & 0 & 0 & -\alpha_4 k_1 & -\alpha_2 k_2 & -\alpha_6 k_3 & 0 & 0 & 0 & 0 \\ 0 & 0 & 0 & 0 & 0 & 0 & 0 & 0 & 0 & 0 & -1 & 0 & 0 & 0 & 0 & 0 & 0 & 1 & 0 & 0 & 0 & 0 & 0 & 0 \\ 0 & 0 & 0 & 0 & 0 & 0 & 0 & 0 & 0 & 0 & 1 & 0 & 0 & 0 & 0 & 0 & 0 & 0 & 0 & 0 & -1 & 0 & 0 \end{bmatrix}$$

$$F = \begin{bmatrix} \alpha_1 & 0 & 0 & \alpha_4 & 0 & 0 \\ \alpha_4 & 0 & 0 & \alpha_2 & 0 & 0 \end{bmatrix}^T$$

The space state differential equations of the suspension system obtained from the vehicle model and the $L_2$ state gain feedback control algorithm are

$$\begin{cases} \dot{X}_P(t) = \Psi X_P(t) + G_P w_1(t) \\ Z = H X_P(t) \end{cases}, \tag{76}$$

where $\Psi = A_P + B_P K$ and $H = E + FK$ are the system state coefficient matrix and output state coefficient matrix obtained after the $L_2$ state gain feedback control transformation of the active suspension, respectively.

The Equation (76) is discretized to obtain

$$\begin{cases} X_P(k+1) = \Phi X_P(k) + \Gamma w_1(k) \\ Z(k) = H X_P(k) \end{cases}, \tag{77}$$

where $\Phi = \Phi(k) = e^{\Psi \tau}$ is the state transition matrix, $\Gamma(k) = \int_0^\tau \Phi(\sigma) d\sigma G_P = \int_0^\tau e^{\Psi \sigma} d\sigma G_P$ is the noise drive matrix and $\tau$ is the time difference, which is 0.01s in this paper.

In the actual working process, the sensor has the problem of random interference during the measurement process. The observed signal often contains observation noise. In order to make the model more realistic, the observed noise $V(k)$ with mean value of 0 and variance matrix of $R$ is introduced into the state equation, then the state equation is

$$\begin{cases} X_P(k+1) = \Phi X_P(k) + \Gamma w_1(k) \\ Z(k) = H X_P(k) + V(k) \end{cases}. \tag{78}$$

### 4.2. Observability of Suspension System

For the established driver-active suspension model, if and only if the driver-active suspension system state variables can be uniquely determined by system model parameters, inputs and outputs, then the system model is observable. Otherwise, the system model is unobservable.

In this paper, the observability is judged by the rank of the observability discriminant matrix based on Kalman filter principle. As the state transition matrix $\Phi$ and the output state coefficient matrix $H$ are assumed to be time-invariant matrices related to system parameters during the sampling period, the observability matrix of the state equation of the driver-active suspension system is defined as

$$M = \begin{bmatrix} H^T & \Phi^T H^T & (\Phi^T)^2 H^T & \cdots & (\Phi^T)^{n-1} H^T \end{bmatrix}, \tag{79}$$

where $n$ is the matrix dimension of the state transition matrix $\Phi$.

There are no non-zero number $k_1, k_2, k_3, \cdots, k_n$ make,

$$k_1 H^T + k_2 \Phi^T(k) H^T + k_3 (\Phi^T(k))^2 H^T + \cdots + k_n (\Phi^T(k))^{n-1} H^T = 0. \tag{80}$$

According to the calculation, the observability matrix of the driver-active suspension model is linearly correlated. The $rank(M) = 24$ and the observability matrix is full rank. Therefore, it is judged that the driver-active suspension model is completely observable, that is, the system state variable at any moment can be uniquely determined.

For any model, an allowable control vector can be found by giving any initial state of the system, and the system is completely controllable if all states of the system are led to the initial state of the state space within a limited time. The control vector of the driver-active suspension model is the active control force generated by the active suspension actuators of the front axle and rear axle. The controllability of the system is proved by the controllability matrix, whose controllability matrix is defined as,

$$S = \begin{bmatrix} \Gamma & \Phi \Gamma & \Phi^2 \Gamma & \cdots & \Phi^{n-1} \Gamma \end{bmatrix}. \tag{81}$$

According to the calculation, the rank of the controllability matrix $S$ is $rank(S) = 24$, and it is full rank. That is, there is a set of control signals $U$, which enables the system to reach the final state $X_{PN}$ at the Nth sampling moment from the initial state $X_{P0}$, so the system is controllable.

*4.3. Kalman Filter of Driver-Active Suspension Model*

The state difference equations of the driver-active suspension model are

$$\begin{cases} X_P(k+1) = \Phi X_P(k) + \Gamma w_1(k) \\ Z(k) = HX_P(k) + V(k) \end{cases}.$$

The premise of using Kalman filtering to estimate the system is that the process noise and measurement noise of the system are unrelated white noise with mean value of 0 and variance matrix of $Q$ and $R$, respectively, and the initial state is not related to process noise and observation noise, that is,

$$\forall k, j, \quad \begin{cases} \delta_{kj} = 1 \quad k = j \\ \delta_{kj} = 0 \quad k \neq j \end{cases},$$

$$E[W(k)] = 0, \; E[V(k)] = 0, \; E\left[W(k)W^T(j)\right] = Q\delta_{kj}, E\left[V(k)V^T(j)\right] = R\delta_{kj},$$

$$E[X(0)] = \mu_0, \; E\left[(X(0) - \mu_0)(X(0) - \mu_0)^T\right] = P_0,$$

where $P_0$ is the initial covariance matrix.

For the Kalman filter algorithm, the first step is to predict the system state at this moment by using the system state at the previous moment,

$$\hat{X}_P(k+1|k) = \Phi(k)\hat{X}_P(k|k), \tag{82}$$

where $\hat{X}_P(k|k)$ is the optimal estimated value of system state at the kth moment, and $\hat{X}_P(k+1|k)$ is the predicted value of the system state values at the $k + 1$st moment according to the system state values at the kth moment.

Then, according to the system error covariance matrix at the previous moment and the process noise variance matrix $Q$ at the present moment, the error covariance at the present moment is predicted,

$$P(k+1|k) = \Phi(k)P(k|k)\Phi(k)^T + \Gamma(k)Q\Gamma(k)_T. \tag{83}$$

Next, the measurement equation is introduced to modify the predicted value of the system state at this moment obtained by Equation (82),

$$\hat{X}_P(k+1|k+1) = \hat{X}_P(k+1|k) + K(k+1)[Z(k) - H\hat{X}_P(k+1|k)] = \hat{X}_P(k+1|k) + K(k+1)\varepsilon(k+1), \tag{84}$$

where $\varepsilon(k+1) = Y(k) - H\hat{X}_P(k+1|k)$ is the difference between the observed value and the prediction value of filtering state, namely, the measurement margin, and $K(k+1)$ is the gain coefficient of Kalman at the current moment, which is related to the output state coefficient matrix, the predicted value of the error covariance, and the variance matrix $R$ of the observed noise. The expression is as follows:

$$K(k+1) = \frac{P(k+1|k)H^T}{HP(k+1|k)H^T + R}, \tag{85}$$

where $R = diag(r_1, r_2, r_3, r_4, r_5, r_6,)$ is the variance matrix of the observed noise. $r_1, r_2, r_3, r_4, r_5$ and $r_6$ represent the observed noise variance of the body acceleration sensor of the front axle, suspension working space sensor of the front axle, the dynamic tire deflection sensor of the front tire, the body acceleration sensor of the rear axle, the

suspension working space sensor of the rear axle, and the dynamic tire deflection sensor of the rear tire, respectively.

Finally, the error covariance matrix is corrected to calculate the system state at the next moment, and the error covariance matrix is updated as

$$P(k+1|k+1) = [I_n - K(k+1)H]P(k+1|k), \tag{86}$$

where $I_n$ is the identity matrix with the same dimension as the system state coefficient matrix.

Equations (82)–(85) are the five core formulas of Kalman filtering algorithm. Wherein, Equations (82) and (83) are the prediction process, and Equations (84)–(86) are the updating process. The relationship between the filter and the system is shown in the Figure 11.

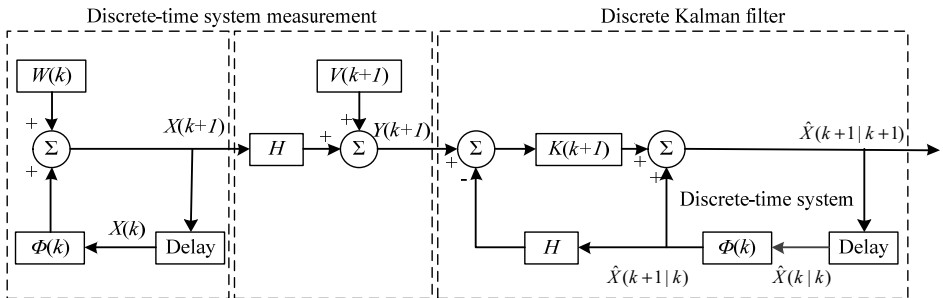

**Figure 11.** Block diagram of discrete Kalman filter.

### *4.4. Adaptive Kalman Filter*

In the actual process, the variance matrix of the measurement noise of the system observer is not constant, and will change with time and road changes, resulting in a decrease in the accuracy of the filtering estimation. The adaptive Kalman filtering algorithm can not only remove the influence of noise changes, but also reduce errors in system modeling.

The adaptive time-varying noise with forgetting factor is

$$r(k) = (1 - d_k)r(k-1) + d_k[Z(k) - HX(k|k-1)], \tag{87}$$

$$R(k) = (1 - d_k)R(k-1) + d_k[\varepsilon(k)\varepsilon^T(k) - HP(k|k-1)H^T], \tag{88}$$

where $d_k = (1 - b)/\left(1 - b^{k+1}\right)$ is the adaptive weighting coefficient, and $0 < b < 1$ is the forgetting factor.

Substituting Equations (87) and (88) into the above linear Kalman filter algorithm, the adaptive Kalman filter equations are obtained,

$$\hat{X}_P(k+1|k) = \Phi\hat{X}_P(k|k), \tag{89}$$

$$R(k) = (1 - d_k)R(k-1) + d_k[\varepsilon(k)\varepsilon^T(k) - HP(k|k-1)H^T], \tag{90}$$

$$K(k+1) = \frac{P(k+1|k)H^T}{HP(k+1|k)H^T + R(k+1)}, \tag{91}$$

$$\varepsilon(k+1) = Z(k) - H\hat{X}_P(k+1|k) - r(k), \tag{92}$$

$$\hat{X}_P(k+1|k+1) == \hat{X}_P(k+1|k) + K(k+1)\varepsilon(k+1), \tag{93}$$

$$P(k+1|k+1) = [I_n - K(k+1)H]P(k+1|k). \tag{94}$$

According to the parameters of vehicle and driver in Tables 1 and 2, the above-mentioned adaptive Kalman filter is used to establish a driver seat-active suspension model. According to the existing literature, the optimal forgetting factor is in the range of 0.8 to 1. In this paper, the step size of 0.01 is used to optimize the forgetting factor.

Due to the difference in magnitude and unit of each state variable, each state variable is normalized to evaluate the filtering effect, and the evaluation index *J* is set as

$$J = \sum_{i=1}^{22} \frac{Mean[\triangle_i(X)]}{Mean[\triangle_{iKF}(X)]},$$ (95)

where *Mean* is the mean value of the filtering deviation of each state variable of the model, $\Delta_i$ is the adaptive Kalman filtering deviation of each state variable of the model, and $\Delta_{iKF}$ is the filtering deviation of each state variable of the linear Kalman filter.

The optimization results are shown in Table 5.

**Table 5.** Forgetting factor search for optimal evaluation index.

| Forgetting factor b | 0.80 | 0.81 | 0.82 | 0.83 | 0.84 | 0.85 | 0.86 | 0.87 | 0.88 | 0.89 |
|---|---|---|---|---|---|---|---|---|---|---|
| J | 0.96 | 0.98 | 0.98 | 0.95 | 1.01 | 1.05 | 1.03 | 1.02 | 1.01 | 1.01 |
| **Forgetting factor b** | 0.90 | 0.91 | 0.92 | 0.93 | 0.94 | 0.95 | 0.96 | 0.97 | 0.98 | 0.99 |
| J | 1 | 0.99 | 0.97 | 0.95 | 0.93 | 0.92 | 0.90 | 0.89 | 0.88 | 0.93 |

It can be seen from Table 5 that the evaluation index J of filtering accuracy changes with the change of forgetting factor b. When the forgetting factor is set to 0.98, the evaluation index J of filtering accuracy is at least 0.88. The smaller the value of the filter accuracy evaluation index J, the smaller the filter error of the adaptive Kalman filter is than the filter error of the linear Kalman filter. In other words, the closer the filtering value is to the true value, the better the filtering effect will be. By comparison, 0.98 is selected as the forgetting factor for adaptive Kalman filter in this paper.

## 5. Test and Simulation

In this paper, a self-made multi-link active suspension scaffolding is used to test and verify the model and control algorithm of this paper. The active suspension actuator consists of a multi-link active suspension and a permanent magnet brushless DC motor. The multi-link active suspension and the test vehicle are shown in Figure 12.

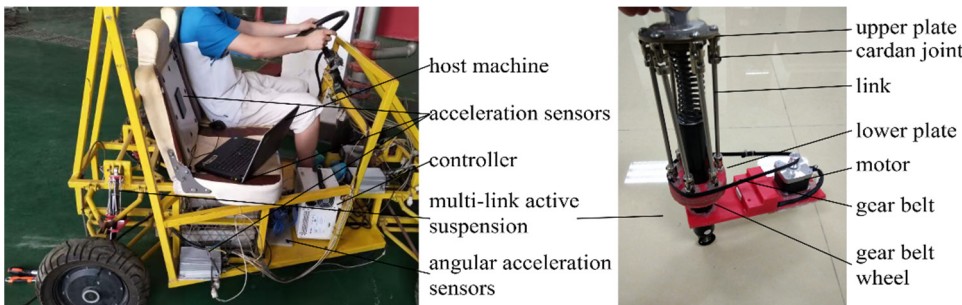

**Figure 12.** Multi-link active suspension and test.

It shown in Figure 12, the multi-link active suspension drives the gear belt tire movement through the control motor, and the links rotates with the gear belt tire to provide active control force to the suspension. The three-way acceleration sensors 327 M are installed on the body, the seat support surface, and the backrest. The angular acceleration sensor is WT61C-232. The parameters of the test vehicle and the active suspension are the same as in Table 1.

The comparison of the test and simulation is shown in Figure 13.

It shown in Figure 13a, the actual and ideal control force curves of the front and rear axle actuators are basically coincident. Through calculation, it can be obtained that the RMS values of the actual active control force of the front axle and rear axle actuators are

243.66 N and 368.64 N, respectively. Compared with the ideal active control force, the difference is only 8.3% and 5.6%. The control effect is ideal.

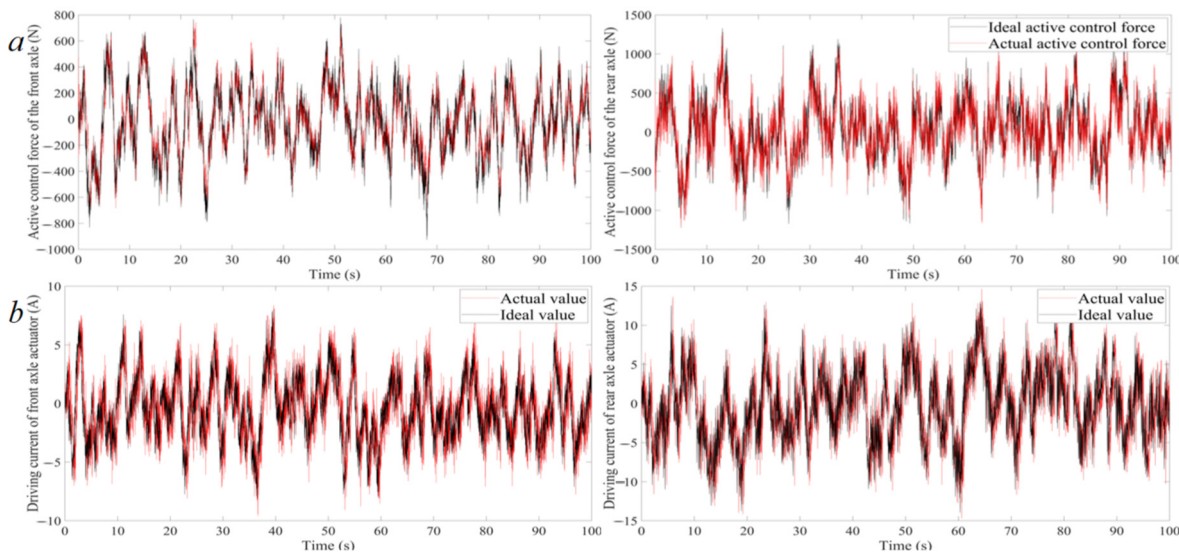

**Figure 13.** Comparison of test and simulation. (**a**) Active control force. (**b**) Drive current of active suspension actuators.

It shown in Figure 13b, the actual driving current of the front and rear actuators is basically the same as the ideal driving current. Through calculation, the RMS value of the actual driving current of the front axle and rear axle actuator can be obtained as 2.7868 A and 4.8211 A respectively. Compared with the ideal active driving current, the difference is only 7.4% and 10.2%.

Figure 14 shows the energy consumption power per unit sprung mass. As can be seen from the figure, in the initial test stage, due to the hysteresis of the motor, there is an error between the ideal value and the real value. However, as time increases, the ideal value curve basically fits the true value curve. After calculation, the true mean value of energy consumption power per unit sprung mass is 0.1624 W/Kg, which is 6.6% different from the ideal value, and the effect is ideal.

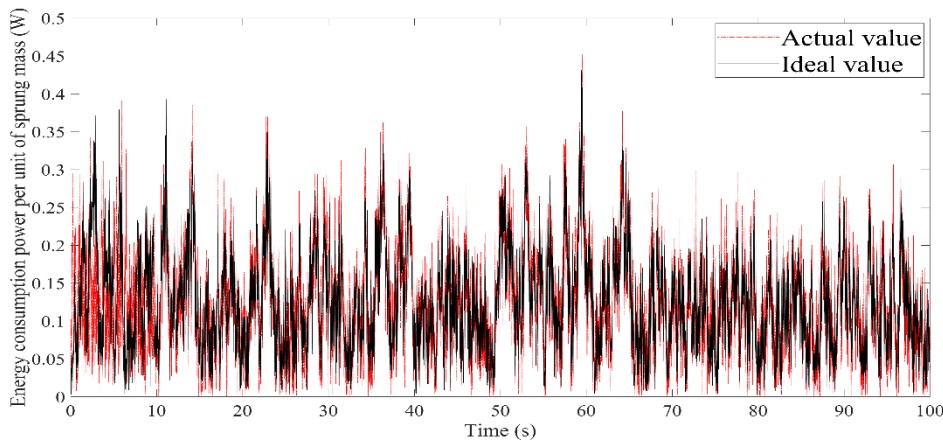

**Figure 14.** Energy consumption power per unit of sprung mass.

It shown in Figure 15a,b that in order to ensure the accuracy of the data, when analyzing the acceleration of the driver's seat support and the pitch angular acceleration of the vehicle body, the intermediate time period of the simulation and the test is intercepted and compared. In the time period of 50–60 s, the RMS values of the acceleration of the driver's seat support in the test and simulation are 0.3407 m·s$^{-2}$ and 0.2989 m·s$^{-2}$,

respectively, and the root mean square values of the pitch angular acceleration of the vehicle body are $0.4018 \, \mathrm{m \cdot s^{-2}}$ and $0.3693 \, \mathrm{m \cdot s^{-2}}$, respectively. Although there are some errors in the simulation and test, the overall trend of change and the magnitude of amplitude are roughly the same, which verifies the accuracy of the above-mentioned theory and simulation.

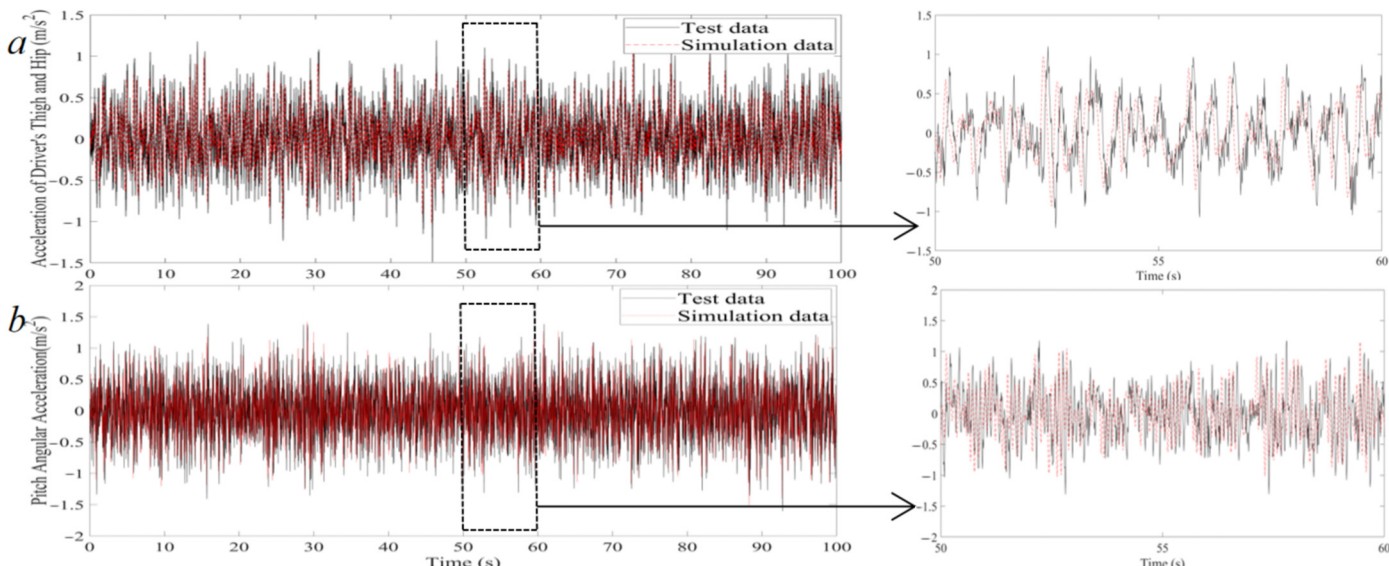

**Figure 15.** Comparison of performance indicators. (**a**) Acceleration of the driver's seat support. (**b**) Pitch angular acceleration.

According to Kalman filter algorithm and adaptive Kalman filter algorithm, the body acceleration of front axle, suspension working space of front axle, dynamic tire deflection of front tire, body acceleration of rear axle, suspension working space of rear axle and dynamic tire deflection of rear tire are selected as observation quantity, and the driver-seat-suspension model is simulated by using B-level road conditions and 20 m/s speed conditions. The results of Kalman filter and adaptive Kalman filter of the observable performance index of the suspension system are shown in Figure 16. The biases between the observed values of the observer, the filtered values of the Kalman filter, the filtered values of the adaptive Kalman filter, and the real values of the system are calculated as shown in Table 6.

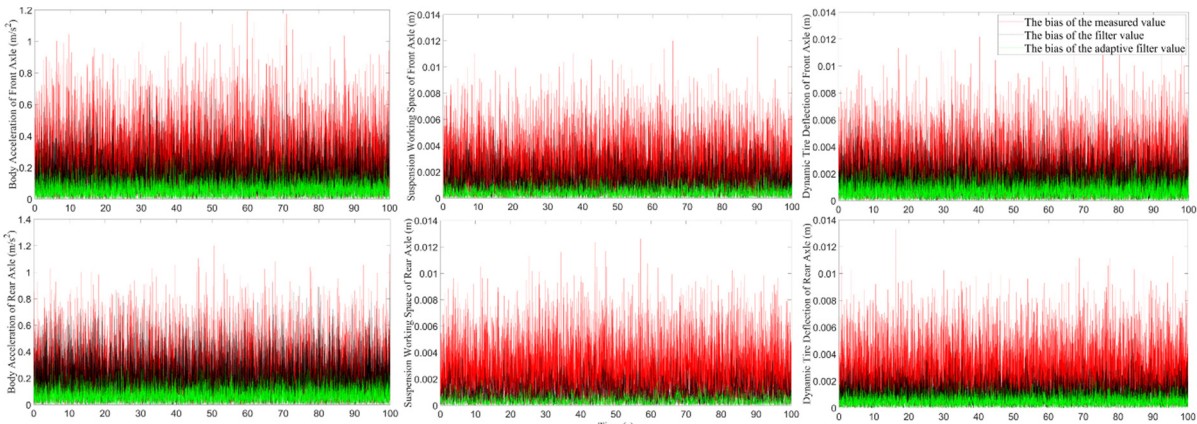

**Figure 16.** Biases of measurable performance indexes.

It shown in Figure 13a, the actual and ideal control force curves of the front and rear axle actuators are basically coincident. Through calculation, it can be obtained that the RMS values of the actual active control force of the front axle and rear axle actuators are

243.66 N and 368.64 N, respectively. Compared with the ideal active control force, the difference is only 8.3% and 5.6%. The control effect is ideal.

**Table 6.** Kalman filter bias values of measurable performance indexes.

| Performance Index | Unit | Observed Bias | | Filter Bias | | Adaptive Filter Bias | |
|---|---|---|---|---|---|---|---|
| | | Mean | Max | Mean | Max | Mean | Max |
| Body acceleration of the front axle $(\ddot{x}_8)$ | m/s$^2$ | 0.1977 | 1.3794 | 0.094 | 0.5977 | 0.057 | 0.2943 |
| Suspension working space of the front axle $(x_8 - x_1)$ | mm | 2.445 | 13.029 | 1.248 | 6.038 | 0.477 | 2.341 |
| Dynamic tire deflection of the front axle $(x_1 - x_{01})$ | mm | 2.445 | 13.142 | 1.350 | 7.751 | 0.698 | 3.352 |
| Body acceleration of the rear axle $(\ddot{x}_9)$ | m/s$^2$ | 0.2543 | 1.2471 | 0.154 | 1.024 | 0.069 | 0.3209 |
| Suspension working space of the rear tire $(x_9 - x_2)$ | mm | 2.448 | 12.458 | 0.975 | 4.579 | 0.409 | 1.934 |
| Dynamic tire deflection of the rear tire $(x_2 - x_{02})$ | mm | 2.672 | 12.176 | 1.136 | 6.074 | 0.499 | 2.423 |

The optimal estimation results of the acceleration of the driver's head and neck, the acceleration of the driver's chest, the acceleration of the driver's waist, the acceleration of the driver's hip and thigh and the pitch angle acceleration are shown in Table 7.

**Table 7.** Kalman filter bias values of immeasurable performance indexes.

| Performance Index | Unit | Filter Bias | | Adaptive Filter Bias | |
|---|---|---|---|---|---|
| | | Mean | Max | Mean | Max |
| Acceleration of the driver's head and neck $(\ddot{x}_7)$ | m/s$^2$ | 0.107 | 0.3756 | 0.043 | 0.1407 |
| Acceleration of the driver's chest $(\ddot{x}_6)$ | m/s$^2$ | 0.095 | 0.3976 | 0.037 | 0.2207 |
| Acceleration of the driver's waist $(\ddot{x}_5)$ | m/s$^2$ | 0.072 | 0.4017 | 0.029 | 0.1675 |
| Acceleration of the driver's hip and thigh $(\ddot{x}_4)$ | m/s$^2$ | 0.084 | 0.4823 | 0.047 | 0.2338 |
| Pitch angle acceleration $(\ddot{\theta})$ | m/s$^2$ | 0.089 | 0.3648 | 0.032 | 0.1991 |

It can be seen from Table 7 that there is no obvious bias between the estimated value of Kalman filter, the estimated value of the adaptive Kalman filter and the actual value of the five state variables, which is not directly observed in the driver seat-active suspension model. The mean biases of adaptive Kalman filter estimation of the five state variables, namely, the acceleration of the driver's head and neck, acceleration of the driver's chest, acceleration of the driver's waist, the acceleration of the driver's hip and thigh, and the pitch angle acceleration, are 0.043, 0.037, 0.029, 0.047, and 0.032 m/s$^2$, respectively. Compared with traditional Kalman filter, the mean filter biases decreased by 59.8%, 61.1%, 59.7%, 44.0%, and 64.0%, respectively. It can be seen that the driver-seat-active suspension filter model established in this paper adopts adaptive Kalman filtering, which can have a good filter estimation effect on the performance indexes that are not directly observed, and can estimate the performance indexes and state variables more accurately, so as to carry out accurate control on the active suspension actuator.

## 6. Conclusions

In order to better improve the driver comfort and vehicle ride comfort, this paper improves the traditional semi-vehicle vibration model, and establishes a 9-degree-of-freedom driver seat-active suspension model. The suspension control algorithm is innovated by using $L_2$ gain feedback control and adaptive Kalman filter algorithm. Through simulation and test, the following conclusions are obtained.

1. The improved 9-degree-of-freedom model can better improve the driving comfort. Compared with the passive suspension, the driving comfort evaluation indexes of the new model are improved by more than 27.2%. Moreover, compared with the traditional active suspension model, all improvements are more than 6.4%.

2. Using $L_2$ gain feedback control and adaptive Kalman filter algorithm to innovate the algorithm. It not only improves the driver comfort, but also comprehensively optimizes the other performance of the vehicle. Compared with the traditional active suspension, the dynamic tire deflection of the front and rear tire decreased by 32.6% and 12.1%, respectively.

3. After the improved model is combined with the innovative algorithm, the actual value of the active control force obtained by the two actuators is basically consistent with the ideal value, the error is less than 8.3%, and the control effect is good.

**Author Contributions:** Conceptualization, H.Y. and J.L. (Jiang Liu); methodology, H.Y.; programming, H.Y.; validation, J.L. (Jiang Liu), M.L. and J.L. (Jianze Liu); writing—original draft preparation, H.Y.; writing—review and editing, H.Y.; supervision, X.Z.; project administration, J.L. (Jiang Liu); funding acquisition, J.L. (Jiang Liu), M.L. and Y.Z. All authors have read and agreed to the published version of the manuscript.

**Funding:** This research is financially supported by the National Natural Science Foundation of China (grant no. 51575288), the National Science Foundation of Shandong Province (grant no. ZR2019MEE072) and the National Science Foundation of Shandong Province (grant no. ZR2019MEE089).

**Institutional Review Board Statement:** Not applicable.

**Informed Consent Statement:** Not applicable.

**Data Availability Statement:** The data used to support the findings of this study are included within the article.

**Acknowledgments:** The authors also express great gratitude to the research team and the editors for their help.

**Conflicts of Interest:** The authors declare no conflict of interest.

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
