# Peer review of "Adaptive Kalman Filter with L2 Feedback Control for Active Suspension Using a Novel 9-DOF Semi-Vehicle Model"

_actuators, doi:10.3390/act10100267_

Round 1

Reviewer 1 Report

The use of first names instead of surnames in the References Section is unacceptable.

The literature review is very general without introducing the authors' thoughts.

The novelty of the work has not been demonstrated.

The authors did not write the purpose of introducing a very complex suspension model with the exact driver model. The authors did not use the potential of the mathematical model derived in the article.

Reviewer 2 Report

The paper has been improved, and the authors answered the reviewer requests. The authors responses address all the reviewer concerns.
In summary, this paper ends up showing critical methodology. Therefore, my recommendation to the journal is to accept the manuscript.

The reviewer kindly suggests revising the References list from line 660 to the end as some are not in MDPI and ACS Style.

Reviewer 3 Report

The paper aims to better improve the driver comfort and vehicle ride comfort. Indeed, the L2 gain feedback control and adaptive Kalman filter algorithm is used for the suspension control.
The paper should be improved by:
- adding some comparisons with existing work on vehicle suspension( see for example:Fault detection for vehicle active suspension systems in finite-frequency domain. IET Control Theory & Applications 13 (3), 387-394, 2019;  A robust predictive control design for nonlinear active suspension systems. Asian Journal of Control 18 (1), 122-132, 2016; ..)
- improving the bibliography on fuzzy control and its application (see for example:Event-Triggered Sliding Mode Control of Networked Fuzzy Systems with Strict Dissipativity. IEEE Transactions on Fuzzy Systems 2021;  H∞ fuzzy proportional integral state feedback controller of photovoltaic systems under asymmetric actuator constraints.Transactions of the Institute of Measurement and Control 43 (1), 34-46, 2020; ...)
- Some typos should be corrected.

Reviewer 4 Report

The manuscript evaluates the performance of an L2 output feedback controller with an adaptive KF to control an active suspension system that is modelled with 9 DOFs including the driver states. The research is interesting, however, the paper suffers from severe issues:

  - Reading the paper is very difficult. It is confusing in many cases:

         - Surprisingly, neither the title nor Abstract mention the keyword active suspension and the context of the research is not clear to the reader till reading the first sentence of the introduction.

        - The title is an unclear mix of L2 feedback and KF. What is the meaning of "L2 Feedback Adaptive Kalman Filter Algorithm"? The problem of the defined driver states is not only the noisy measurement but is the fact that even measuring the states could be practically impossible.

        - What is the difference between sections 4.1 and 4.2?

       - Please explain the derivation of (28-31) from (27) in more detail. What is f0?

       - (35) is not notation-wisely correct. There are some misusing of terminologies. For example, U and W are not matrices but are vectors.

       - (32) and (38) indicate that the number of states is 24, however, the values under (53) imply the same coordinate as 26. Please explain. Also, how are (52) and the matrices under (53) related?  Generally speaking, the structure of section 3.3 and the relationship between the provided materials are not clear.

       - The reasons for adding section 3.3 are not clear. 

       - The number of indicated DoFs in Figure 2 is 10 which does not match the claimed 9-DoF model.  x10 is not independent and depends on x8 and x9.

 - The Abstract of the paper needs a thorough modification. The sentences are not clear. For example, it says "this paper proposes an improved 9-DOF model." however it does not say the model of what.

  - English wording needs a thorough modification. There are long paragraphs with sentences that are not linked properly.

   - How the provided simulation results in 3.4 illustrate the superiority of the developed 9 DoF model + L2 feedback controller against other models. What do you mean by 'traditional LQG model"? LQG is a controller, not a model. What is the structure, system model and objectives of the studied LQG? the provided figures are not readable and need a further process to represent in a better way. Why is 3.4 not part of section 5?

Round 2

Reviewer 1 Report

The novelty of the work is based on introducing a complex 9-DOF model. However, the authors do not use the potential of this model. I do not find this control in a real application. The paper is a compilation of other work and does not bring anything new to the controlled suspension area.

Figure 1 is still wrong. Authors should use the summation node. 

The authors strongly argue that using the 9-DOF model is necessary, but they do not use all state variables for control. The model adopted is excessive and not used for control (see quality indicator). The variables x3 x5 x6 are not taken into account.

The efficiency of the active suspension control mainly depends on the energy supplied to the system but not on the adopted model.

The authors do not use limitation (saturation) for the control signal.

The set of references is wrongly matched to the presented area of science.

The References format (name instead Family Name) still looks terrible.

I do not recommend this paper to be purplish in a scientific journal.

Reviewer 3 Report

The revision manuscript is satisfactory.  

Reviewer 4 Report

N/A

Author Response

This manuscript is a resubmission of an earlier submission. The following is a list of the peer review reports and author responses from that submission.

Round 1

Reviewer 1 Report

The mathematical modeling of car suspension system has been quite mature in the published literature.

The theory developed in Section 4 is not novel which is standard in the textbooks of optimal estimation.

According to the schematic diagram illustrated in Fig. 1 there should be an active controller but no control term appeared in the Kaman filtering model in Section 4.

Equation (33) is incorrect for the matrix case.

Equation (66) given for the Kalman gain is incorrect for the matrix case.

Figure 6b shows a real multi-link active suspension system, which is interesting, but only simulation analysis of the model is presented.

Reviewer 2 Report

Thank you for the opportunity to review this manuscript.

The authors present an analytical study of vehicles suspension controllers. Firstly, the authors establish a discrete driver-seat-active suspension semi-vehicle model, solve the ten linear matrix inequalities, and obtain the optimal feedback gain of the model L2 state matrix. This is an exciting problem, and the designed solution for different components involved, improving efficiency, and strengthening high low-speed torque, has value to the community. The solutions achieved in the paper will be helpful in the future field’s evolution. The authors present an extensive review of the bibliography showing ways to introduce and study the problem. The starting citations in the paper could be improved by adding some other cases that could benefit the present study, i.e. adding some of the follows where simulation/emulation has been used to validate vehicle subsystems in Model/Software/Hardware in the loop of MHEVs:

Forstinger, M., Bauer, R., Hofer, A., & Rossegger, W. (2016). Multivariable control of a test bed for differential gears. Control engineering practice, 57, 18-28.

Scaradozzi, D., & Fanesi, M. (2019). Advanced Control Strategies to Improve Nonlinear Automotive Dynamical Systems Consumption. Axioms, 8(4), 123.

Dzierżek, S. (2000). Experiment-based modeling of cylindrical rubber bushings for the simulation of wheel suspension dynamic behavior. SAE transactions, 78-85.

Zeng, X., & Wang, J. (2015). A parallel hybrid electric vehicle energy management strategy using stochastic model predictive control with road grade preview. IEEE Transactions on Control Systems Technology, 23(6), 2416-2423.

The authors present a number of simulations to validate their hypothesis on the field, and the preliminary results of the model and the designed modelling system are very promising. 

The reviewer kindly suggests expanding considerations about computing power and time.

The RMS values of the performance indexes of the two different models are calculated respectively, as shown in Table 4. The reviewer kindly suggests adding the mean value in order to validate the general amplitude of RMS better.

In summary, this paper ends up showing that build a modular, lightweight model to simulate and control the behaviour of active suspension systems. Therefore, my recommendation is to accept the manuscript with minor revisions.

Reviewer 3 Report

Comments to the Authors

The authors present a L2 feedback Kalman filter algorithm to design the suspension controller. They establish a semi-vehicle 9 degree-of-freedom driver-active suspension model for simulation. They show several results of the new model and the traditional LQG active suspension and compare them.

Some of my personal opinions are here:

  • Most of the papers cited in the introduction part are to some extent old. It is suggested to add more new researches about active suspension. The follow papers can be referred to: DOI: 10.1016/j.jsv.2020.115171
  • In the summary (line 17-21), there is no clearly explanation of which method is used to compare with the authors’ method.
  • In line 84-86, it might be better to replace the word ‘part’ with ‘section’.
  • There is a lack of explanation of some parameters. For example, the explanation of the equation (27) is insufficient (i.e. v, f), and the G0 is not explained in the equation (29).
  • In line 171-172, the serial numbers of the equations (35) (36) are wrong, and there are some similar errors such as the serial number of the table 1 on line 207. Besides, the equation (42) exists a mistake about the transpose.
  • Some parts of the writing can be written better (i.e. line 130 243-259…) so that the meaning of the sentences are clearly to the reader, and there exist several grammatical mistakes in the article (i.e. line 171 190 446…).

Reviewer 4 Report

The presented parameters are not related to reality. The authors do not specify what type of suspension they are considering. The motion ratios of passive and active elements were not taken into account. The values of the parameters of passive elements have not been reduced (converted) to the vertical component. It is a huge disadvantage.

There is no information related to external energy demand for the active system. The efficiency of the active vibration control depends mainly on the supplied energy. The authors did not take into account the dynamics of the actuator.

The authors did not discuss the determination of the Y matrix coefficients (equation 47). These are very crucial issues determining controller efficiency.

The time courses of the control are not shown. Without evaluating the range of this signal, it is impossible to reliable judge the controller’s efficiency.

PSD unit is wrong in Figure 5.

Wymuszenie normowe od drogi jest uzyskane jako niezależne sygnały losowe z nałożonym filtrem charakteryzującym drogę. W rzeczywistości tak nie jest sygnały te powinny być skorelowane co wynika z rozstawu osi. To znacząco wpływa na syntezę filtru Kalmana.

The excitation is simulated as independent Gaussian white noise signals with the applied filter characterising the road irregularities. It is not the case, and these signals should be correlated due to the wheelbase. The actual delay between signals x01 and x02  (or w1, w2 ) is significant affects the synthesis of the Kalman filter.

The proposed mathematical model - used to controller synthesises - includes the hip and thigh, waist, chest and head were built. Nevertheless, only the hip and thigh and neck were analysed. All of these organ’s movements of the seat-driver were not included in the performance indexes.

The authors did not present the novelty of the work. It should be done against the background of the state of the Art. The LQG controller and the Kalman filter are well known and are often used to synthesise active suspension systems. The submitted manuscript is a reproductive-engineering work and does not bring anything new to the field. Therefore, I do not recommend this work for publication in Actuators Journal.

Round 2

Reviewer 1 Report

NA

Reviewer 3 Report

The revised mansucript can be accpted for publication.

Reviewer 4 Report

Reply 1: In the suspension model adopted by you, all forces act in the vertical direction (Figure 2). However, in the suspension of the Ford Granada described by you, this is not the case. To transform all forces generated by the passive elements to the vertical direction, you should take the motion and force ratios into account. The similar transformation you should do in the active system. The dynamic parameters, frequency and/or time constants of the active element could be helpful.

Reply 2: Yes, you have all signals, and you can calculate the energy used by the active element to control vibration per each unit of mass in the presence of the specified disturbances. This is when the paper will have a scientific and comparative value. One can design any control algorithm; the question is only at what energy cost. This information must be complete.

Reply 3: Fascinating, the weights coefficients have changed so significantly and the controller continues to do the same. Could use hard proof.

Reply 4: Rather, I was thinking about the control signal coming directly from the controller not being converted into force.    

Reply 5: well done         

Reply 6: looks better

Reply 7: satisfying

Reply 8: satisfying